# Connecting Thompson Sampling and UCB: Towards More Efficient Trade-offs Between Privacy and Regret

**Bingshan Hu** [1]   **Zhiming Huang** [2]   **Tianyue H. Zhang** [3 4]   **Mathias Lécuyer** [1]   **Nidhi Hegde** [5 6]

## Abstract

We address differentially private stochastic bandit problems by leveraging Thompson Sampling with Gaussian priors and Gaussian differential privacy (GDP). We propose DP-TS-UCB, a novel parametrized private algorithm that enables trading off privacy and regret. DP-TS-UCB satisfies $\tilde{O}\left(T^{0.25(1-\alpha)}\right)$-GDP and achieves $O\left(K \ln^{\alpha+1}(T)/\Delta\right)$ regret bounds, where $K$ is the number of arms, $\Delta$ is the sub-optimality gap, $T$ is the learning horizon, and $\alpha \in [0, 1]$ controls the trade-off between privacy and regret. Theoretically, DP-TS-UCB relies on anti-concentration bounds for the Gaussian distributions, linking the exploration mechanisms of Thompson Sampling and Upper Confidence Bound, which may be of independent research interest.

## 1. Introduction

This paper studies differentially private stochastic bandit problems previously studied in Mishra & Thakurta (2015); Hu et al. (2021); Hu & Hegde (2022); Azize & Basu (2022); Ou et al. (2024). In a classical stochastic bandit problem, we have a fixed arm set $[K]$. Each arm $i$ is associated with a fixed but unknown reward distribution $p_i$ with mean reward $\mu_i$. In each round, a learning agent pulls an arm and obtains a random reward that is distributed according to the reward distribution associated with the pulled arm. The goal of the learning agent is to pull arms sequentially to accumulate as much reward as possible over a finite number of $T$ rounds. Since the pulled arm in each round may not always be the optimal one, *regret*, defined as the expected cumulative

loss between the highest mean reward and the earned mean reward, is used to measure the performance of the algorithm used by the learning agent to make decisions.

Low-regret bandit algorithms should leverage past information to inform future decisions, as previous observations reveal which arms have the potential to yield higher rewards. However, due to privacy concerns, the learning agent may not be allowed to directly use past information to make decisions. For example, a hospital collects health data from patients participating in clinical trials over time to learn the side effects of some newly developed treatments. To comply with privacy regulations, the hospital is required to publish scientific findings in a differentially private manner, as the sequentially collected data from patients carries sensitive information from individuals. The framework of differential privacy (DP) (Dwork et al., 2014) is widely accepted to preserve the privacy of individuals whose data have been used for data analysis. Differentially private learning algorithms bound the privacy loss, the amount of information that an external observer can infer about individuals.

DP is commonly achieved by adding noise to summary statistics computed based on the collected data. Therefore, to solve a private bandit problem, the learning agent has to navigate two trade-offs. The first one is the *fundamental trade-off between exploitation and exploration* due to bandit feedback: in each round, the learning agent can only focus on either exploitation (pulling arms seemingly promising to attain reward) or exploration (pulling arms helpful to learn the unknown mean rewards and reduce uncertainty). The second one is the *trade-off between privacy and regret* due to the DP noise: adding more noise enhances privacy, but it reduces data estimation accuracy and weakens regret guarantees.

There are two main strategies to design (non-private) stochastic bandit algorithms that efficiently balance exploitation and exploration: Upper Confidence Bound (UCB) (Auer et al., 2002) and Thompson Sampling (Agrawal & Goyal, 2017; Kaufmann et al., 2012b). Both enjoy good theoretical regret guarantees and competitive empirical performance. UCB is inspired by the principle of optimism in the face of uncertainty, adding deterministic bonus terms to the empirical estimates based on their uncertainty to achieve

---

[1]Department of Computer Science, University of British Columbia, Canada. [2]Department of Computer Science, University of Victoria, Canada. [3]Université de Montréal, Canada. [4]Mila – Quebec AI Institute, Canada. [5]Department of Computing Science, University of Alberta, Canada. [6]Alberta Machine Intelligence Instribute (Amii), Canada. Correspondence to: Bingshan Hu <bingshanhu3@gmail.com>.

*Proceedings of the 42nd International Conference on Machine Learning*, Vancouver, Canada. PMLR 267, 2025. Copyright 2025 by the author(s).

exploration. Thompson Sampling is inspired by Bayesian learning, using the idea of sampling mean reward models from posterior distributions (e.g., Gaussian distributions) that model the unknown mean rewards of each arm. The procedure of sampling mean reward models can be viewed as adding random bonus terms to the empirical estimates.

The design of the existing private stochastic bandit algorithms (Sajed & Sheffet, 2019; Hu et al., 2021; Azize & Basu, 2022; Hu & Hegde, 2022) follows the framework of adding calibrated noise to the empirical estimates first to achieve privacy. Then, the learning agent makes decisions based on noisy estimates, which can be viewed as post-processing that preserves DP guarantees. Since both Thompson Sampling and DP algorithms rely on adding noise to the empirical estimates, it is natural to wonder whether the existing Thompson Sampling-based algorithms offer some level of privacy at no additional cost, without compromising any regret guarantees.

Very recently, Ou et al. (2024) show that Thompson Sampling with Gaussian priors (Agrawal & Goyal, 2017) (we rename it as TS-Gaussian in this work) without any modification is indeed DP by leveraging Gaussian privacy mechanism (adding Gaussian noise to the collected data (Dwork et al., 2014)) and the notion of Gaussian differential privacy (GDP) (Dong et al., 2022). They show that TS-Gaussian is $O(\sqrt{T})$-GDP. However, this privacy guarantee is not tight due to the fact that *TS-Gaussian has to sample a mean reward model from a data-dependent Gaussian distribution for each arm in each round to achieve exploration*. Each sampled Gaussian mean reward model implies the injection of some Gaussian noise into the empirical estimate, and sampling in total $T$ Gaussian mean reward models for each arm provides a privacy guarantee in the order of $\sqrt{T}$.

In this paper, we propose a novel private bandit algorithm, DP-TS-UCB (presented in Algorithm 1), which does not require sampling a Gaussian mean reward model in each round, and is hence more efficient at trading off privacy and regret. Theoretically, DP-TS-UCB uncovers the connection between exploration mechanisms in TS-Gaussian and UCB1 (Auer et al., 2002), which may be of independent interest.

Our proposed algorithm builds upon the insight that, for each arm $i$, the Gaussian distribution that models the mean reward of arm $i$ can only change when arm $i$ is pulled, as a new pull of arm $i$ indicates the arrival of new data associated with arm $i$. In other words, the Gaussian distribution stays the same in all rounds between two consecutive pulls of arm $i$. Based on this insight, to avoid unnecessary Gaussian sampling, which increases privacy loss, DP-TS-UCB **sets a budget $\phi$ for the number of Gaussian mean reward models that are allowed to draw from a Gaussian distribution.** Among all the rounds between two consecutive pulls of arm $i$, DP-TS-UCB can only draw a Gaussian mean

reward model in each of the first $\phi$ rounds. If arm $i$ is still not pulled after $\phi$ rounds, DP-TS-UCB reuses the highest model value among the previously sampled $\phi$ Gaussian mean reward models in the remaining rounds until arm $i$ is pulled again. Figure 1 presents a concrete example of how DP-TS-UCB works.

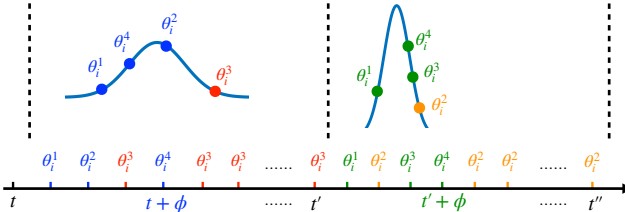

*Figure 1.* **Cap the number of mean reward models sampled from a Gaussian distribution.** Assume arm $i$ is pulled in rounds $t$, $t'$ and $t''$, and $\phi = 4$. In each of the rounds $t+1, \ldots, t+h, \ldots, t+\phi$, DP-TS-UCB samples a Gaussian mean reward model $\theta_i^h$ and uses it in the learning for arm $i$. In each of the rounds $t+\phi+1, t+\phi+2, \ldots, t'$, DP-TS-UCB reuses the highest model value $\theta_i^3 = \max_{h \in [\phi]} \theta_i^h$ among the previously sampled $\phi$ mean reward models in the learning for arm $i$. Once a new Gaussian distribution is available (the Gaussian distribution located on the right side), DP-TS-UCB is allowed to draw $\phi$ Gaussian mean reward models again in each of the rounds $t'+1, t'+2, \ldots, t'+\phi$.

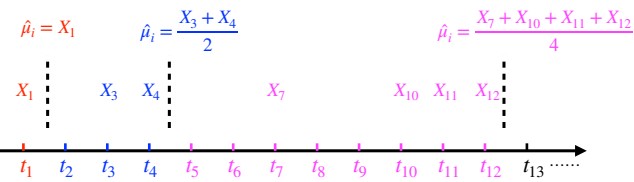

*Figure 2.* **Arm-specific epoch structure.** The dashed lines partition rounds from $t_1$ to $t_{12}$ into three epochs. Assume arm $i$ is pulled in round $t_1$, then we compute its empirical mean as $\hat{\mu}_i = X_1$ at the end of round $t_1$ and arm $i$'s first epoch ends in round $t_1$. If arm $i$ is pulled in rounds $t_3, t_4$ again, then we compute its empirical mean as $\hat{\mu}_i = (X_3 + X_4)/2$ at the end of round $t_4$ and arm $i$'s second epoch ends in round $t_4$. It is important to note that arm $i$'s empirical mean will not be updated at the end of round $t_3$ even though it is pulled in round $t_3$.

To have a tight privacy guarantee, in addition to capping the number of Gaussian mean reward models, we also need to limit the number of times that a revealed observation can be used when computing empirical estimates. Similar to Sajed & Sheffet (2019); Hu et al. (2021); Azize & Basu (2022); Hu & Hegde (2022), we use an **arm-specific epoch structure** to process the revealed observations. As already discussed in these works, using this structure is the key to designing good private online learning algorithms. The key idea of this structure is to update the empirical estimate using the most recent $2^r$ observations, where $r \geq 0$. Figure 2 illustrates this structure for the first three epochs.

**Preview of results.** DP-TS-UCB uses an input parameter $\alpha \in [0, 1]$ to control the trade-off between privacy and regret, and the choice of $\phi = O(T^{0.5(1-\alpha)} \ln^{0.5(3-\alpha)}(T))$ depends on both $\alpha$ and the learning horizon $T$. Our technical Lemma 4.1 shows that this choice of $\phi$ ensures sufficient exploration, that is, giving enough optimism, for the rounds when sampling new Gaussian mean reward models is not allowed. DP-TS-UCB is $\tilde{O}(T^{0.25(1-\alpha)})$-GDP (Theorem 4.4) and achieves $\sum_{i:\Delta_i>0} O(\ln(\phi T \Delta_i^2) \ln^\alpha(T)/\Delta_i)$ regret bounds (Theorem 4.2), where $\Delta_i$ is the mean reward gap between the optimal arm and a sub-optimal arm $i$. For the case where $\alpha = 0$, DP-TS-UCB enjoys the optimal $\sum_{i:\Delta_i>0} O(\ln(\phi T \Delta_i^2)/\Delta_i)$ regret bounds and satisfies $\tilde{O}(T^{0.25})$-GDP, which improves the previous $O(\sqrt{T})$-GDP guarantee significantly. For the case where $\alpha = 1$, DP-TS-UCB satisfies constant $\tilde{O}(1)$-GDP and achieves $\sum_{i:\Delta_i>0} O(\ln(\phi T \Delta_i^2) \ln(T)/\Delta_i)$ regret bounds.

## 2. Learning Problem

In this section, we first present the learning problem of stochastic bandits and then we provide key knowledge related to differentially private online learning.

### 2.1. Stochastic Bandits

In a classical stochastic bandit problem, we have a fixed arm set $[K]$ of size $K$, and each arm $i \in [K]$ is associated with a fixed but unknown reward distribution $p_i$ with $[0, 1]$ support. Let $\mu_i$ denote the mean of reward distribution $p_i$. Without loss of generality, we assume that the first arm is the unique optimal arm, i.e., $\mu_1 > \mu_i$ for all $i \neq 1$. Let $\Delta_i := \mu_1 - \mu_i$ denote the mean reward gap. The learning protocol is in each round $t$, a reward vector $X_t := (X_1(t), X_2(t), \ldots, X_K(t))$ is generated, where each $X_i(t) \sim p_i$. Simultaneously, the learning agent pulls an arm $i_t \in [K]$. At the end of the round, the learning agent receives a reward $X_{i_t}(t)$. The goal of the learning agent is to pull arms sequentially to maximize the cumulative reward over $T$ rounds, or equivalently, minimize the *(pseudo)-regret*, defined as

$$\mathcal{R}(T) = T \cdot \mu_1 - \mathbb{E}\left[\sum_{t=1}^{T} \mu_{i_t}\right], \tag{1}$$

where the expectation is taken over the pulled arm $i_t$. The regret measures the expected cumulative mean reward loss between always pulling the optimal arm and the learning agent's actual pulled arms.

### 2.2. Differential Privacy

Our DP definition in the context of online learning follows the one used in Dwork et al. (2014); Sajed & Sheffet (2019); Hu et al. (2021); Hu & Hegde (2022); Azize & Basu (2022); Ou et al. (2024). Let $X_{1:t} := (X_1, X_2, \ldots, X_t)$ collect all

the reward vectors up to round $t$. Let $X'_{1:t}$ be a neighbouring sequence of $X_{1:t}$ which differs in at most one reward vector, say, in some round $\tau \leq t$.

**Definition 2.1** (DP in online learning). An online learning algorithm $\mathcal{A}$ is $(\varepsilon, \delta)$-DP if for any two neighbouring reward sequences $X_{1:T}$ and $X'_{1:T}$, for any decision set $\mathcal{D}_{1:t} \subseteq [K]^t$, we have $\mathbb{P}\{\mathcal{A}(X_{1:t}) \in \mathcal{D}_{1:t}\} \leq e^\varepsilon \cdot \mathbb{P}\{\mathcal{A}(X'_{1:t}) \in \mathcal{D}_{1:t}\} + \delta$ holds for all $t \leq T$ simultaneously.

Like Ou et al. (2024), we also perform our analysis using Gaussian differential privacy (GDP) (Dong et al., 2022), which is well suited to analyzing the composition of Gaussian mechanisms. We then translate the GDP guarantee to the classical $(\varepsilon, \delta)$-DP guarantee by using the *duality* between GDP and DP (Theorem 2.4). Indeed, Dong et al. (2022) show that GDP can be viewed as the primal privacy representation with its dual being an infinite collection of $(\varepsilon, \delta)$-DP guarantees.

To introduce GDP, we first need to define trade-off functions:

**Definition 2.2** (Trade-off function (Dong et al., 2022)). For any two probability distributions $P$ and $Q$ on the same space, define the trade-off function $T(P, Q) : [0, 1] \rightarrow [0, 1]$ as $T(P, Q)(x) = \inf_\psi \{\beta_\psi : \alpha_\psi \leq x\}$, where $\alpha_\psi = \mathbb{E}_P[\psi]$, $\beta_\psi = 1 - \mathbb{E}_Q[\psi]$, and the infimum is taken over all measurable rejection rules $\psi \in [0, 1]$.

Let $\Phi$ denote the cumulative distribution function (CDF) of the standard normal distribution $\mathcal{N}(0, 1)$. To define GDP in the context of online learning, for any $\eta \geq 0$, we let $G_\eta(x) := T(\mathcal{N}(0, 1), \mathcal{N}(\eta, 1))(x) = \Phi(\Phi^{-1}(1 - x) - \eta)$ denote the trade-off function of two normal distributions.

**Definition 2.3** ($\eta$-GDP in online learning). A randomized online learning algorithm $\mathcal{A}$ is $\eta$-GDP if for any two reward vector sequences $X_{1:T}$ and $X'_{1:T}$ differing in at most one vector, we have $T(\mathcal{A}(X_{1:t}), \mathcal{A}(X'_{1:t}))(x) \geq G_\eta(x)$ holds for all $x \in [0, 1]$ and $t \leq T$ simultaneously.

For easier comparison, we use the following theorem to convert an $\eta$-GDP guarantee to $(\varepsilon, \delta)$-DP guarantees:

**Theorem 2.4** (Primal to dual (Dong et al., 2022)). *A randomized algorithm is $\eta$-GDP if and only if it is $(\varepsilon, \delta(\varepsilon))$-DP for all $\varepsilon \geq 0$, where*

$$\delta(\varepsilon) = \Phi\left(-\frac{\varepsilon}{\eta} + \frac{\eta}{2}\right) - e^\varepsilon \Phi\left(-\frac{\varepsilon}{\eta} - \frac{\eta}{2}\right).$$

**Remark.** Fix any $\varepsilon \geq 0$. We can also view $\delta(\varepsilon) = \Phi\left(-\frac{\varepsilon}{\eta} + \frac{\eta}{2}\right) - e^\varepsilon \Phi\left(-\frac{\varepsilon}{\eta} - \frac{\eta}{2}\right)$ as an increasing function of $\eta$. This means, for a fixed $\varepsilon$, the smaller the GDP parameter $\eta$ is, the smaller the $\delta(\varepsilon)$ is after the translation.

## 3. Related Work

There is a vast amount of literature on (non-private) stochastic bandit algorithms. We split them based on UCB-based versus Thompson Sampling-based, i.e., deterministic versus randomized exploration. Then, we discuss the most relevant algorithms for private stochastic bandits.

UCB-based algorithms (Auer et al., 2002; Audibert et al., 2007; Garivier & Cappé, 2011; Kaufmann et al., 2012a; Lattimore, 2018) usually conduct exploration in a deterministic way. The key idea is to construct confidence intervals centred on the empirical estimates. Then, the learning agent makes decisions based on the upper bounds of the confidence intervals. The widths of the confidence intervals control the exploration level. Thompson Sampling-based algorithms (Agrawal & Goyal, 2017; Kaufmann et al., 2012b; Bian & Jun, 2022; Jin et al., 2021; 2022; 2023) conduct exploration in a randomized way. The key idea is to use a sequence of well-chosen data-dependent distributions to model each arm's mean reward. Then, the learning agent makes decisions by sampling random mean reward models from these distributions. The spread of the data-dependent distributions controls the exploration level. In addition to the aforementioned algorithms, we also have DMED (Honda & Takemura, 2010), IMED (Honda & Takemura, 2015), elimination-style algorithm (Auer & Ortner, 2010), Non-parametric TS (Riou & Honda, 2020), and Generic Dirichlet Sampling (Baudry et al., 2021). All these algorithms enjoy either $\sum_{i:\Delta_i>0} O(\ln(T)/\Delta_i)$ or $\sum_{i:\Delta_i>0} O(\ln(T)\Delta_i/\text{KL}(\mu_i, \mu_i + \Delta_i))$ problem-dependent regret bounds, where $\text{KL}(a,b)$ denotes the KL-divergence between two Bernoulli distributions with parameters $a, b \in (0,1)$.

Sajed & Sheffet (2019); Azize & Basu (2022); Hu et al. (2021) developed optimal $(\varepsilon, 0)$-DP stochastic bandit algorithms by first adding calibrated Laplace noise to the empirical estimates to ensure $(\varepsilon, 0)$-DP. Then, eliminating arms and constructing data-dependent distributions based on noisy estimates can be viewed as post-processing which do not hurt privacy. Although Hu & Hegde (2022) proposed a private Thompson Sampling-based algorithm, it still follows the above recipe without leveraging the inherent randomness present in Thompson Sampling for privacy.

Ou et al. (2024) connected Thompson Sampling with Gaussian priors (we rename it as TS-Gaussian) (Agrawal & Goyal, 2017) to the Gaussian privacy mechanism (Dwork et al., 2014) and Gaussian differential privacy (Dong et al., 2022). The idea of TS-Gaussian is to use $\mathcal{N}(\hat{\mu}_{i,n_i}, 1/n_i)$ to model arm $i$'s mean reward, i.e., the mean of reward distribution $p_i$. The centre of the Gaussian distribution $\hat{\mu}_{i,n_i}$ is the empirical average of $n_i$ observations that are i.i.d. according to $p_i$. To decide which arm to pull, for each arm $i$ in each round, the learning agent samples a

Gaussian mean reward model $\theta_i \sim \mathcal{N}(\hat{\mu}_{i,n_i}, 1/n_i)$. The learning agent pulls the arm with the highest mean reward model value. Ou et al. (2024) showed that TS-Gaussian satisfies $\sqrt{0.5T}$-GDP, before translating this GDP guarantee to $(\varepsilon, \delta)$-DP guarantees with Theorem 2.4. Since there is no modification to the original algorithm, the optimal $\sum_{i:\Delta_i>0} O(\ln(T\Delta_i^2)/\Delta_i)$ problem-dependent regret bounds and the near-optimal $O(\sqrt{KT\ln(K)})$ worst-case regret bounds are preserved. Ou et al. (2024) also proposed Modified Thompson Sampling with Gaussian priors (we rename it as M-TS-Gaussian), which enables a privacy and regret trade-off. Compared to TS-Gaussian, the modifications are pre-pulling each arm $b$ times and scaling the variance of the Gaussian distribution as $c/n_i$. They proved that M-TS-Gaussian satisfies $\sqrt{T/(c(b+1))}$-GDP, and achieves $bK + \sum_{i:\Delta_i>0} O(c\ln(T\Delta_i^2)/\Delta_i)$ problem-dependent regret bounds and $bK + O(c\sqrt{KT\ln K})$ worst-case regret bounds. Table 1 summarizes the theoretical results of TS-Gaussian and M-TS-Gaussian with different choices of $b, c$.

The order of $\sqrt{T}$-GDP guarantee from TS-Gaussian and M-TS-Gaussian may not be tight when $T$ is large. There are two reasons resulting in this loose privacy guarantee: (1) sampling a Gaussian mean reward model in each round for each arm injects too much noise; (2) repeatedly using the same observation to compute the empirical estimates creates too much privacy loss. In this work, we propose DP-TS-UCB, a novel private algorithm that does not require sampling a Gaussian mean reward model in each round for each arm. The intuition is that once we are confident some arm is sub-optimal, we do not need to further explore it. To avoid using the same observation to compute the empirical estimates, we use the arm-specific epoch structure devised by Hu et al. (2021); Azize & Basu (2022); Hu & Hegde (2022) to process the obtained observations. Using this structure ensures that each observation can only be used at most once for computing empirical estimates.

Regarding lower bounds with a finite learning horizon $T$ for differentially private stochastic bandits, lower bounds exist under the classical $(\varepsilon, \delta)$-DP notion. Shariff & Sheffet (2018) established $\Omega(\sum_{i:\Delta_i>0} \ln(T)/\Delta_i + K\ln(T)/\varepsilon)$ problem-dependent regret lower bound and Azize & Basu (2022) established an $\Omega(\sqrt{KT} + K/\varepsilon)$ minimax regret lower bound for $(\varepsilon, 0)$-DP. Wang & Zhu (2024) established an $\Omega(\sum_{i:\Delta_i>0} \ln(T)/\Delta_i + \frac{K}{\varepsilon}\ln\frac{(e^\varepsilon-1)T+\delta T}{(e^\varepsilon-1)+\delta T})$ problem-dependent regret lower bound for $(\varepsilon, \delta)$-DP. In this work, we do not provide any new lower bounds. Our theoretical results are compatible with these established lower bounds.

## 4. DP-TS-UCB

We present DP-TS-UCB and then provide its regret (Theorem 4.2) and privacy (Theorems 4.4 and 4.6) guarantees.

**Algorithm 1** DP-TS-UCB

1: **Input:** trade-off parameter $\alpha \in [0, 1]$, learning horizon $T$, and budget $\phi = c_0 T^{0.5(1-\alpha)} \ln^{0.5(3-\alpha)}(T)$.
2: **Initialization:** (1) pull each arm $i$ once to initialize $n_i$ and $\hat{\mu}_{i,n_i}$, (2) set arm-specific epoch index $r_i \leftarrow 1$ and the number of unprocessed observations $O_i \leftarrow 0$, (3) set remaining Gaussian sampling budget $h_i \leftarrow \phi$ and the highest Gaussian mean reward model $\text{MAX}_i \leftarrow 0$.
3: **for** $t = K+1, K+2, \ldots, T$ **do**
4:    **for** $i \in [K]$ **do**
5:       **if** $h_i \geq 1$ **then**
6:          Set $\theta_i(t) \leftarrow \theta_{i,n_i}^{(h_i)}$, where $\theta_{i,n_i}^{(h_i)} \sim \mathcal{N}\left(\hat{\mu}_{i,n_i}, \frac{\ln^\alpha(T)}{n_i}\right)$ %Mandatory TS-Gaussian
7:          Set $h_i \leftarrow h_i - 1, \text{MAX}_i \leftarrow \max\{\text{MAX}_i, \theta_{i,n_i}^{(h_i)}\}$
8:       **else**
9:          Set $\theta_i(t) \leftarrow \text{MAX}_i$ %Optional UCB
10:       **end if**
11:    **end for**
12:    Pull arm $i_t \in \arg\max_{i \in [K]} \theta_i(t)$, observe $X_{i_t}(t)$, and set $O_{i_t} \leftarrow O_{i_t} + 1$
13:    **if** $O_{i_t} = 2^{r_{i_t}}$ **then**
14:       Compute $\hat{\mu}_{i_t, n_{i_t}}$, where $n_{i_t} = 2^{r_{i_t}}$
15:       Reset $h_{i_t} \leftarrow \phi, \text{MAX}_{i_t} \leftarrow 0$
16:       Set $r_{i_t} \leftarrow r_{i_t} + 1$ and reset $O_{i_t} \leftarrow 0$.
17:    **end if**
18: **end for**

### 4.1. DP-TS-UCB Algorithm

Algorithm 1 presents the pseudo-code of DP-TS-UCB. Let $c_0 = \sqrt{2\pi e}$. We input trade-off parameter $\alpha \in [0, 1]$ and learning horizon $T$, and then we compute the sampling budget $\phi = c_0 T^{0.5(1-\alpha)} \ln^{0.5(3-\alpha)}(T)$. Let $n_i(t-1)$ denote the number of observations that are used to compute the empirical estimate $\hat{\mu}_{i,n_i(t-1)}$ at the end of round $t-1$.

**Initialize learning algorithm (Line 2).** There are several steps to initialize the learning algorithm. (1) We pull each arm $i \in [K]$ once to initialize each arm's empirical mean $\hat{\mu}_{i,n_i}$. Since the decisions in these rounds do not rely on any data, we do not have any privacy concerns. (2) As we use the *arm-specific epoch structure* (Figure 2 describes the key ideas of this structure) to process observations, we use $r_i$ to track arm $i$'s epoch progress and use $O_i$ to count the number of unprocessed observations in epoch $r_i$. We initialize $r_i = 1$ and $O_i = 0$. (3) Since we *can only draw at most $\phi$ mean reward models from each Gaussian distribution*, we use $h_i$ to count the remaining Gaussian sampling budget at the end of round $t-1$, and $\text{MAX}_i$ to track the maximum value among these $\phi$ Gaussian mean reward models. Initially, we set $h_i = \phi$ and $\text{MAX}_i = 0$.

**Decide learning models (Line 4 to Line 11).** Let $\theta_i(t)$

denote arm $i$'s learning model in round $t \geq K + 1$. Each $\theta_i(t)$ can either be *a new Gaussian mean reward model* or *some Gaussian mean reward model already used before*. To decide which case fits arm $i$ in round $t$, we check the value of $h_i$ to see whether drawing a new Gaussian mean reward from $\mathcal{N}\left(\hat{\mu}_{i,n_i(t-1)}, \ln^\alpha(T)/n_i(t-1)\right)$ is allowed: if $h_i \geq 1$, we sample a new mean reward model $\theta_{i,n_i}^{(h_i)} \sim \mathcal{N}\left(\hat{\mu}_{i,n_i(t-1)}, \ln^\alpha(T)/n_i(t-1)\right)$ and use it in the learning, i.e., $\theta_i(t) = \theta_{i,n_i}^{(h_i)}$; if $h_i = 0$, we use $\theta_i(t) = \text{MAX}_i = \max_{h_i \in [\phi]} \theta_{i,n_i}^{(h_i)}$ in the learning as we have all $\theta_{i,n_i}^{(1)}, \theta_{i,n_i}^{(2)}, \ldots, \theta_{i,n_i}^{(\phi)}$ in hand already.

Our technical Lemma 4.1 below shows that the highest mean reward model $\text{MAX}_i$ is analogous to the upper confidence bound in UCB1 (Auer et al., 2002). The usage of $\text{MAX}_i$ ensures sufficient exploration for the rounds when sampling new Gaussian mean reward models is not allowed. We can view DP-TS-UCB as a two-phase algorithm with a mandatory TS-Gaussian phase and an optional UCB phase. Note that DP-TS-UCB itself does not explicitly construct upper confidence bounds; $\text{MAX}_i$ itself behaves like the upper confidence bound of arm $i$ in UCB1 in terms of achieving exploration.

**Lemma 4.1.** *Fix any observation number $s \geq 1$ and let $\theta_{i,s}^{(1)}, \ldots, \theta_{i,s}^{(\phi)}$ be i.i.d. according to $\mathcal{N}\left(\hat{\mu}_{i,s}, \ln^\alpha(T)/s\right)$. We have $\mathbb{P}\left\{\max_{h \in [\phi]} \theta_{i,s}^{(h)} \geq \mu_i\right\} \geq 1 - O(1/T)$.*

**Make a decision and collect data (Line 12).** With all learning models $\theta_i(t)$ in hand, the learning agent pulls the arm $i_t \in \arg\max_{i \in [K]} \theta_i(t)$ with the highest model value, observes $X_{i_t}(t)$ and increments the unprocessed observation counter $O_{i_t}$ by one.

**Process collected data (Line 13 to Line 17).** To control the number of times any observation can be used when computing the empirical mean, we only update the empirical mean of the pulled arm $i_t$ when the number of unprocessed observations $O_{i_t} = 2^{r_{i_t}}$. After the update, we reset $h_{i_t}, O_{i_t}$ and $\text{MAX}_{i_t}$, and increment the epoch progress $r_{i_t}$ by one.

**Remark on Algorithm 1.** We use data collected in epoch $r_i - 1$ in a differentially private manner to guide the future data collection in epoch $r_i$. We have a mandatory TS-Gaussian phase where drawing Gaussian mean reward models is allowed and an optional UCB phase where the agent can only reuse the best Gaussian mean reward model in the mandatory TS-Gaussian phase. Separating all the rounds belonging to epoch $r_i$ into two possible phases controls the cumulative injected noise (and privacy loss) regardless of the epoch length.

### 4.2. Regret Analysis of DP-TS-UCB

In this section, we provide a regret analysis of Algorithm 1.

**Theorem 4.2.** *The problem-dependent regret bound of DP-TS-UCB with trade-off parameter $\alpha \in [0,1]$ is*

$$\sum_{i:\Delta_i>0} O\left(\frac{\ln\left(T^{0.5(3-\alpha)}\Delta_i^2\right)\ln^\alpha(T)}{\Delta_i} + \frac{(3-\alpha)\ln\ln(T)\cdot\ln^\alpha(T)}{\Delta_i}\right).$$

*The worst-case regret bound of DP-TS-UCB with trade-off parameter $\alpha \in [0,1]$ is $O(\sqrt{KT}\ln^{0.5(1+\alpha)}(T))$.*

Theorem 4.2 gives the following corollary immediately.

**Corollary 4.3.** *DP-TS-UCB with trade-off parameter $\alpha = 0$ achieves $\sum_{i:\Delta_i>0} O\left(\ln\left(T^{1.5}\Delta_i^2\right)/\Delta_i\right) + O\left(\ln\ln(T)/\Delta_i\right)$ problem-dependent regret bounds and $O(\sqrt{KT\ln(T)})$ worst-case regret bounds.*

**Discussion.** DP-TS-UCB with parameter $\alpha = 0$ can be viewed as a problem-dependent optimal bandit algorithm with theoretical guarantees lying between TS-Gaussian (Agrawal & Goyal, 2017) and UCB1 (Auer et al., 2002); the $\sum_{i:\Delta_i>0} O\left(\ln\left(T^{1.5}\Delta_i^2\right)/\Delta_i\right) + O\left(\ln\ln(T)/\Delta_i\right)$ bound is better than the $\sum_{i:\Delta_i>0} O\left(\ln(T)/\Delta_i\right)$ bound of UCB1, but it is slightly worse than the $\sum_{i:\Delta_i>0} O\left(\ln(T\Delta_i^2)/\Delta_i\right)$ bound of TS-Gaussian. DP-TS-UCB with parameter $\alpha = 0$ is not optimal in terms of regret guarantees, but it offers a constant GDP guarantee (see Corollary 4.5 in Section 4.3).

We sketch the proof of Theorem 4.2. The full proof is deferred to Appendix D. Since DP-TS-UCB lies in between TS-Gaussian and UCB1, the regret analysis includes key ingredients extracted from both algorithms.

*Proof sketch of Theorem 4.2.* Fix a sub-optimal arm $i$. Let $L_i = O\left(\ln(\phi T\Delta_i^2)\ln^\alpha(T)/\Delta_i^2\right)$ indicate the number of observations needed to sufficiently observe sub-optimal arm $i$. We know that the total regret accumulated from arm $i$ before $i$ is sufficiently observed is at most $L_i \cdot \Delta_i$. By tuning $L_i$ properly, for all the rounds when arm $i$ is observed sufficiently, the regret accumulated from arm $i$ can be upper bounded by

$$\sum_{t=K+1}^{T} \mathbb{P}\left\{i_t = i, \theta_i(t) \le \mu_i + 0.5\Delta_i\right\} \quad , \quad (2)$$

where $\theta_i(t)$ can either be a fresh Gaussian mean reward model (TS-Gaussian phase, Line 6) or the highest Gaussian mean model used before (UCB phase, Line 9).

We further decompose (2) based on whether the optimal arm 1 is in TS-Gaussian phase (Line 6) or UCB phase (Line 9). Define $\mathcal{T}_1(t)$ as the event that the optimal arm 1 uses a fresh Gaussian mean reward model in round $t$, i.e., in TS-Gaussian phase, and let $\overline{\mathcal{T}_1(t)}$ denote the complement. We have (2) decomposed as

$$\sum_{t=K+1}^{T} \mathbb{P}\left\{i_t = i, \theta_i(t) \le \mu_i + 0.5\Delta_i, \mathcal{T}_1(t)\right\}$$
$$+ \sum_{t=K+1}^{T} \mathbb{P}\left\{i_t = i, \theta_i(t) \le \mu_i + 0.5\Delta_i, \overline{\mathcal{T}_1(t)}\right\} \quad , \quad (3)$$

where, generally, the first term will use the regret analysis of TS-Gaussian in Agrawal & Goyal (2017) and the second term will use Lemma 4.1. In Appendix D, we present an improved analysis of TS-Gaussian and show that the regret of the first term is at most $O\left(\ln(\phi T\Delta_i^2)\ln^\alpha(T)/\Delta_i\right)$. The second term uses a union bound and Lemma 4.1, and is at most $\sum_{t=K+1}^{T} \mathbb{P}\left\{i_t = i, \theta_i(t) \le \mu_i + 0.5\Delta_i, \overline{\mathcal{T}_1(t)}\right\} \le \sum_{t=K+1}^{T} \mathbb{P}\left\{\theta_1(t) \le \mu_1, \overline{\mathcal{T}_1(t)}\right\} \le O(\ln(T))$. $\square$

### 4.3. Privacy Analysis of DP-TS-UCB

This section provides the privacy analysis of Algorithm 1.

**Theorem 4.4.** *DP-TS-UCB with trade-off parameter $\alpha \in [0,1]$ satisfies $\sqrt{2c_0 T^{0.5(1-\alpha)}\ln^{1.5(1-\alpha)}(T)}$-GDP.*

Theorem 4.4 gives the following corollary immediately.

**Corollary 4.5.** *DP-TS-UCB with trade-off parameter $\alpha = 0$ satisfies $O\left(T^{0.25}\ln^{0.75}(T)\right)$-GDP; DP-TS-UCB with trade-off parameter $\alpha = 1$ satisfies $O(1)$-GDP.*

**Discussion.** Together, Theorem 4.2 (regret guarantees) and Theorem 4.4 (privacy guarantees) show that DP-TS-UCB is able to trade off privacy and regret. The privacy guarantee improves with the increase of trade-off parameter $\alpha$, at the cost of suffering more regret.

Table 1 summarizes privacy and regret guarantees of TS-Gaussian (Agrawal & Goyal, 2017), M-TS-Gaussian (Ou et al., 2024), and DP-TS-UCB. From the results, even for the worst case, i.e., $\alpha = 0$, DP-TS-UCB is still $\tilde{O}\left(T^{0.25}\right)$-GDP, which could be much better than the $O(\sqrt{T})$-GDP guarantee of TS-Gaussian. Since DP-TS-UCB with $\alpha = 1$ achieves a constant GDP guarantee, increasing learning horizon $T$ does not increase privacy cost. M-TS-Gaussian pre-pulls each arm $b$ times and uses $c/n_i$ as the Gaussian variance. Generally, it achieves $bK + \sum_{i:\Delta_i>0} O(c\log(T\Delta_i^2)/\Delta_i)$ regret bounds and satisfies $\sqrt{T/(c(b+1))}$-GDP. By tuning $b, c = O(\ln^\alpha(T))$, M-TS-Gaussian achieves $\sum_{i:\Delta_i>0} O(\ln^\alpha(T)\log(T\Delta_i^2)/\Delta_i)$ regret bounds (almost the same as DP-TS-UCB's regret bounds), but satisfying $O(\sqrt{T}/\ln^\alpha(T))$-GDP guarantees, which could be much worse than the $\tilde{O}\left(T^{0.25}\right)$-GDP guarantees of DP-TS-UCB. By tuning $b, c = O(T^\gamma)$, where $\gamma > 0$, M-TS-Gaussian achieves $\sum_{i:\Delta_i>0} O(T^\gamma \log(T\Delta_i^2)/\Delta_i)$ regret bounds and satisfies $O(\sqrt{T^{1-2\gamma}})$-GDP. Although the GDP guarantee is improved to be in the order of $\sqrt{T^{1-2\gamma}}$, the regret bound may be worse than DP-TS-UCB's bounds due to the existence of the $T^\gamma$ term. For example, when setting $\gamma = 0.25$, M-TS-Gaussian is $O(T^{0.25})$-GDP, but it has a $\sum_{i:\Delta_i>0} O(T^{0.25}\log(T\Delta_i^2)/\Delta_i)$ regret bound, which will not be problem-dependent optimal.

Since the classical $(\varepsilon, \delta)$-DP notion is more interpretable, we translate GDP presented in Theorem 4.4 into

*Table 1.* Summary of privacy and regret guarantees

| | Regret bounds | GDP guarantees |
|---|---|---|
| TS-G (Agrawal & Goyal, 2017) | $O\left(K\ln(T\Delta^2)/\Delta\right)$ | $O(T^{0.5})$ |
| M-TS-G (Ou et al., 2024) | $bK + O(cK\ln(T\Delta^2)/\Delta)$ | $O(\sqrt{T/(c(b+1))})$ |
| M-TS-G (tune $b$,$c$ = $O(T^\gamma),\gamma>0$) | $O(KT^\gamma\ln(T\Delta^2)/\Delta)$ | $O(T^{0.5-\gamma})$ |
| M-TS-G (tune $b$,$c$ = $O(\ln^\alpha(T))$) | $O(K\ln^\alpha(T)\ln(T\Delta^2)/\Delta)$ | $O(T^{0.5}/\ln^\alpha(T))$ |
| DP-TS-UCB (Algorithm 1) | $O(K\ln\left(T^{0.5(3-\alpha)}\Delta^2\right)\ln^\alpha(T)/\Delta + K\ln\ln(T)\ln^\alpha(T)/\Delta)$ | $O(T^{0.25(1-\alpha)}\ln^{0.75(1-\alpha)}(T))$ |
| DP-TS-UCB (tune $\alpha = 0$) | $O(K\ln\left(T^{1.5}\Delta^2\right)/\Delta + K\ln\ln(T)/\Delta)$ | $\tilde{O}\left(T^{0.25}\right)$ |
| DP-TS-UCB (tune $\alpha = 1$) | $O(K\ln\left(T\Delta^2\right)\ln(T)/\Delta) + K\ln\ln(T)\ln(T)/\Delta)$ | $O(1)$ |

$(\varepsilon, \delta)$-DP guarantees by using Theorem 2.4.

**Theorem 4.6.** *DP-TS-UCB is $(\varepsilon, \delta(\varepsilon))$-DP for all $\varepsilon \geq 0$, where $\delta(\varepsilon) = \Phi\left(-\frac{\varepsilon}{\sqrt{2\phi}} + \frac{\sqrt{2\phi}}{2}\right) - e^\varepsilon \cdot \Phi\left(-\frac{\varepsilon}{\sqrt{2\phi}} - \frac{\sqrt{2\phi}}{2}\right)$, where $\phi = c_0 T^{0.5(1-\alpha)}\ln^{0.5(3-\alpha)}(T)$.*

*Proof.* Directly using Theorem 2.4 concludes the proof. □

The proof for Theorem 4.4 relies on the following composition theorem and post-processing theorem of GDP.

**Theorem 4.7** (GDP composition (Dong et al., 2022)). *The $m$-fold composition of $\eta_j$-GDP mechanisms is $\sqrt{\eta_1^2 + \ldots + \eta_m^2}$-GDP.*

**Theorem 4.8** (GDP Post-processing (Dong et al., 2022)). *If a mechanism $\mathcal{A}$ is $\eta$-GDP, its post-processing is also $\eta$-GDP.*

*Proof of Theorem 4.4.* Fix any two neighbouring reward sequences $X_{1:T} = (X_1, \ldots, X_\tau \ldots, X_T)$ and $X'_{1:T} = (X_1, \ldots, X'_\tau, \ldots X_T)$, where the complete reward vector in round $\tau$ is changed. Under the bandit feedback model, this change only impacts the empirical mean of the arm pulled in round $\tau$, that is arm $i_\tau$. Name $i_\tau = j$: based on the arm-specific epoch structure (Figure 2), the observation $X_j(\tau)$ will only be used once for computing the empirical mean of arm $j$ at the end of some future round, which is the last round of some epoch $r_j - 1$ associated with arm $j$.

We have one Gaussian distribution constructed using $X_j(\tau)$ at the beginning of epoch $r_j$. If arm $j$ only has the mandatory TS-Gaussian phase in epoch $r_j$, we draw at most $\phi$ Gaussian mean reward models from that constructed Gaussian distribution. From Lemma 5 of Ou et al. (2024), we know DP-TS-UCB is $\sqrt{1/\ln^\alpha(T)}$-GDP in each round in the mandatory TS-Gaussian phase. From Theorem 4.7, we know the GDP composition over at most $\phi$ rounds is $\sqrt{\phi/\ln^\alpha(T)}$-GDP. Note that $X_j(\tau)$ will not be used to construct Gaussian distributions starting from epoch $r_j + 1$ to the end of learning due to the usage of arm-specific epoch structure, i.e., we abandon $X_j(\tau)$ at the end of epoch $r_j$.

If arm $j$ has both the mandatory TS-Gaussian phase and the optional UCB phase in epoch $r_j$, for the mandatory TS-Gaussian phase, DP-TS-UCB is $\sqrt{\phi/\ln^\alpha(T)}$-GDP; for

the optional UCB phase, DP-TS-UCB is also $\sqrt{\phi/\ln^\alpha(T)}$-GDP, as by post-processing Theorem 4.8, the maximum MAX$_j$ of $\phi$ Gaussian mean reward models is $\sqrt{\phi/\ln^\alpha(T)}$-GDP. Composing the privacy guarantees in these two phases concludes the proof. □

# 5. Experimental Results

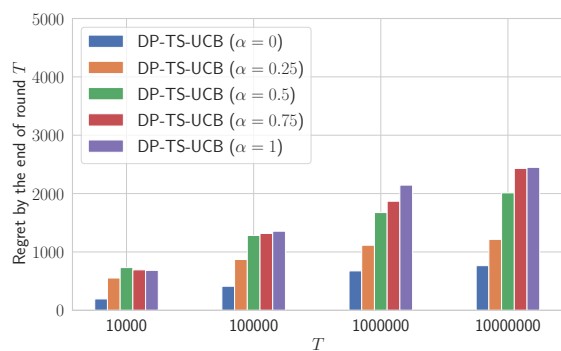

(a) The impact of learning horizon $T$ and trade-off parameter $\alpha$ on the regret by the end of round $T$.

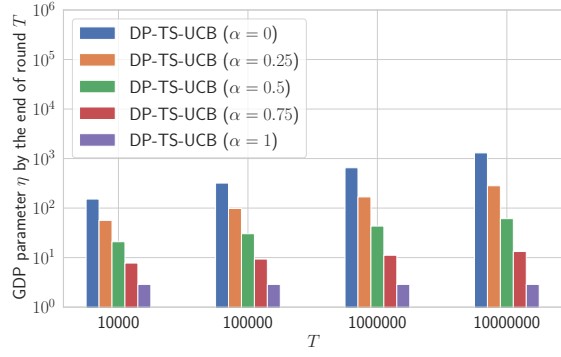

(b) The impact of learning horizon $T$ and trade-off parameter $\alpha$ on the GDP parameter $\eta$ by the end of round $T$.

*Figure 3.* DP-TS-UCB's privacy vs regret with different $\alpha$ and $T$.

The setup consists of five arms with Bernoulli rewards. We set the mean rewards as $[0.95, 0.75, 0.55, 0.35, 0.15]$. We first analyze DP-TS-UCB's privacy and regret across different values of $\alpha$ and $T$. Then, we compare DP-TS-UCB with M-TS-Gaussian (Ou et al., 2024) from two perspectives:

(1) **Privacy cost under equal regret**; (2) **Regret under equal privacy guarantee**. We also compare with $(\varepsilon, 0)$-DP algorithms, including DP-SE (Sajed & Sheffet, 2019), Anytime-Lazy-UCB (Hu et al., 2021), and Lazy-DP-TS (Hu & Hegde, 2022) for $\varepsilon = 0.5$, which can be found in Appendix E.2. All the experimental results are an average of 20 independent runs on a MacBook Pro with M1 Max and 32GB RAM.

### 5.1. Privacy and Empirical Regret of DP-TS-UCB with Different Values of $\alpha$ and $T$

The performance of DP-TS-UCB in terms of the privacy guarantees and regret across different values of $\alpha$ and time horizons $T$ are shown in Figure 3. The results reveal a trade-off between regret minimization and privacy preservation: increasing $\alpha$ leads to a stronger privacy guarantee, reflected in a lower GDP parameter $\eta$, but at the cost of higher regret. However, when $\alpha = 1$, the privacy guarantee becomes constant, meaning that increasing $T$ no longer deteriorates the privacy protection of DP-TS-UCB.

### 5.2. Privacy and Empirical Regret Comparison under the Same Theoretical Regret Bound

Since DP-TS-UCB with parameter $\alpha$ and M-TS-Gaussian with parameters $b = 0$, $c = 5\ln^\alpha(T)$) share the same theoretical regret bound, we now present empirical regret and privacy guarantees for different values of $\alpha = \{0, 0.25, 0.5, 0.75, 1\}$. We set $T = 10^6$. Figure 4(a) shows that DP-TS-UCB incurs lower empirical regret than M-TS-Gaussian, whereas Figure 4(b) shows that DP-TS-UCB achieves better privacy.

### 5.3. Empirical Regret Comparison under the Same Privacy Guarantee

M-TS-Gaussian satisfies a $\sqrt{T/(c(b+1))}$-GDP guarantee, while DP-TS-UCB satisfies $\sqrt{2c_0 T^{0.5(1-\alpha)} \ln^{1.5(1-\alpha)}(T)}$-GDP. Thus, we let $c = \sqrt{\frac{1}{2c_0(b+1)} T^{0.5(1+\alpha)} \ln^{-1.5(1-\alpha)} T}$ for any $b$ of M-TS-Gaussian to ensure the same privacy guarantees as DP-TS-UCB. We compare their empirical regret over $T = 10^6$ rounds under two privacy settings determined by $\alpha$ for both algorithms. For each $\alpha$, we select $b$ from $\{0, 1, 500, 1000, 2000, 5000, 100000\}$ to minimize regret of M-TS-Gaussian (see Appendix E.1).

$\sqrt{2c_0 T^{0.5} \ln^{1.5} T}$-**GDP Guarantee** ($\alpha = 0$). The optimal M-TS-Gaussian parameters are $b = 1$ and $c = 1.18$. As shown in Figure 5(a), M-TS-Gaussian slightly outperforms DP-TS-UCB, but the empirical regret gap is small.

$\sqrt{2c_0}$-**GDP Guarantee** ($\alpha = 1$). For this setting, the best M-TS-Gaussian parameters are $b = 2000$ and $c = 60.46$. However, DP-TS-UCB achieves lower regret, significantly

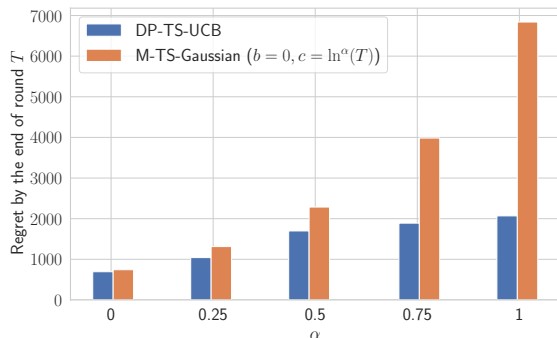

(a) Regret by the end of round $T$ of DP-TS-UCB and M-TS-Gaussian with different parameters.

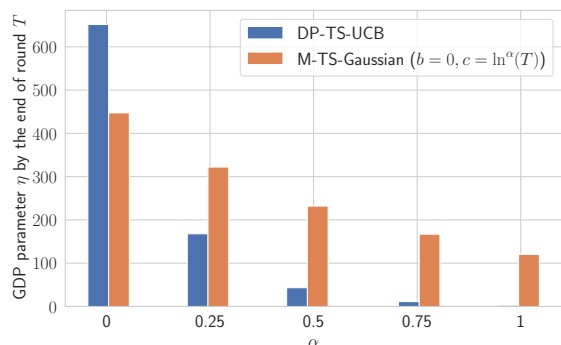

(b) GDP parameter $\eta$ by the end of round $T$ of DP-TS-UCB and M-TS-Gaussian with different parameters.

*Figure 4.* The performance of DP-TS-UCB and M-TS-Gaussian under the same theoretical regret bound.

outperforming M-TS-Gaussian, as shown in Figure 5(b).

## 6. Conclusion

This paper presents a novel private stochastic bandit algorithm DP-TS-UCB (Algorithm 1) by leveraging the connection between exploration mechanisms in TS-Gaussian and UCB1. We first show that DP-TS-UCB satisfies $\tilde{O}(T^{0.25(1-\alpha)})$-GDP and then we translate this GDP guarantee to the classical $(\varepsilon, \delta)$-DP guarantees by using duality between these two privacy notions. Corollary 4.3 and Corollary 4.5 show that DP-TS-UCB with parameter $\alpha = 0$ achieves the optimal $O(K \ln(T)/\Delta)$ problem-dependent regret bounds and the near-optimal $O(\sqrt{KT \ln T})$ worst-case regret bounds, and satisfies $\tilde{O}(T^{0.25})$-GDP. This privacy guarantee could be much better than the $O(\sqrt{T})$-GDP guarantees achieved by TS-Gaussian and M-TS-Gaussian of Ou et al. (2024). We conjecture that our privacy improvement is at the cost of the anytime property of the learning algorithm and the worst-case regret bounds. Note that both TS-Gaussian and M-TS-Gaussian are anytime and achieve $O(\sqrt{KT \ln K})$ worst-case regret bounds, whereas our DP-

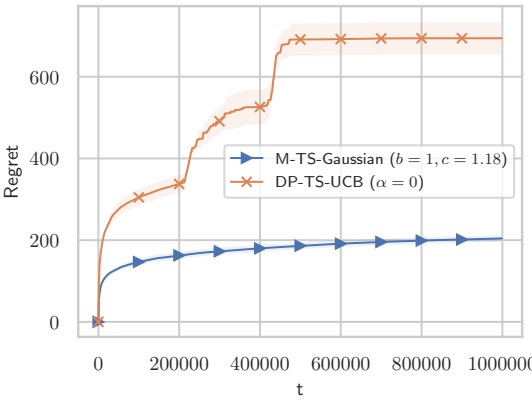

(a) Empirical regret for $O(\sqrt{T^{0.5}\ln^{1.5}T})$ -GDP ($\alpha = 0$).

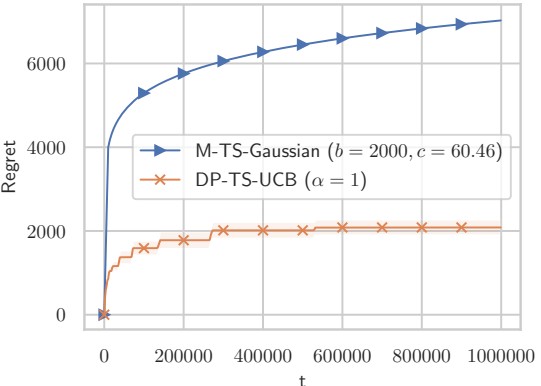

(b) Empirical regret for $\sqrt{2c_0}$-GDP ($\alpha = 1$).

*Figure 5.* The regret of DP-TS-UCB and M-TS-Gaussian under the same privacy guarantee with $\alpha = 0$ and $\alpha = 1$.

TS-UCB is not anytime and achieves only $O(\sqrt{KT\ln T})$ worst-case regret bounds. If we know the maximum mean reward gap $\Delta_{\max} = \max_{i\in[K]}\Delta_i$ in advance, by slightly modifying the theoretical analysis, we know a better choice of $\phi$ should be the one depending on $\Delta_{\max}$. Tuning $\phi$ that depends on $\Delta_{\max}$ will provide problem-dependent GDP guarantees. This intuition motivates us to develop private algorithms that achieve problem-dependent GDP guarantees as the main future work.

## Acknowledgements

Bingshan Hu is grateful for the funding support from the Natural Sciences and Engineering Resource Council of Canada (NSERC), the Canada CIFAR AI chairs program, and the UBC Data Science Institute. Zhiming Huang would like to acknowledge funding support from NSERC and the British Columbia Graduate Scholarship. Tianyue H. Zhang is grateful for the support from Canada CIFAR AI chairs program and Samsung Electronics Co., Limited. Mathias Lécuyer is grateful for the support of NSERC with reference number RGPIN-2022-04469. Nidhi Hegde would like to acknowledge funding support from the Canada CIFAR AI Chairs program.

## Impact Statement

Privacy-preserving sequential decision-making is important in modern interactive machine learning systems, particularly in bandit learning and its general variant reinforcement learning (RL). Our work contributes to this field by proposing a novel differentially private bandit algorithm that connects classical algorithms in the RL community and DP community. Understanding the interplay between decision-making algorithms like Thompson Sampling, and privacy mechanisms and notions is fundamental to advance the deployment of RL algorithms using sensitive data.

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

## A. Useful facts

*Fact* A.1. For any $T > e^3$, for any $\alpha \in [0, 1]$, we have $\ln^{1-\alpha}(T) \leq (1 - \alpha)\ln(T) + 1$.

*Proof.* Let function $f(\alpha) = (1 - \alpha)\ln(T) + 1 - \ln^{1-\alpha}(T)$, where variable $\alpha \in [0, 1]$. Then, we have $f'(\alpha) = -\ln(T) + \ln^{1-\alpha}(T)\ln(\ln(T))$. It is not hard to verify that $f'\left(\frac{\ln(\ln(\ln(T)))}{\ln(\ln(T))}\right) = 0$. The fact that $f'(\alpha) \geq 0$ when $\alpha \in \left[0, \frac{\ln(\ln(\ln(T)))}{\ln(\ln(T))}\right]$ gives $f(\alpha) \geq f(0) = 1 > 0$ for any $\alpha \in \left[0, \frac{\ln(\ln(\ln(T)))}{\ln(\ln(T))}\right]$. Similarly, the fact that $f'(\alpha) \leq 0$ when $\alpha \in \left[\frac{\ln(\ln(\ln(T)))}{\ln(\ln(T))}, 1\right]$ gives $f(\alpha) \geq f(1) = 0$ for any $\alpha \in \left[\frac{\ln(\ln(\ln(T)))}{\ln(\ln(T))}, 1\right]$. Therefore, we have $f(\alpha) \geq 0$ for any $\alpha \in [0, 1]$. □

*Fact* A.2 (Hoeffding's inequality). Let $X_1, X_2, \ldots, X_n$ be $n$ independent random variables with support $[0, 1]$. Let $\mu_{1:n} = \frac{1}{n}\sum_{i=1}^{n} X_i$. Then, for any $a > 0$, we have $\mathbb{P}\{|\mu_{1:n} - \mathbb{E}[\mu_{1:n}]| \geq a\} \leq 2e^{-2na^2}$.

*Fact* A.3 (Concentration and anti-concentration bounds of Gaussian distributions). For a Gaussian distributed random variable $Z$ with mean $\mu$ and variance $\sigma^2$, for any $z > 0$, we have

$$\mathbb{P}\{Z > \mu + z\sigma\} \leq \tfrac{1}{2}e^{-\frac{z^2}{2}}, \quad \mathbb{P}\{Z < \mu - z\sigma\} \leq \tfrac{1}{2}e^{-\frac{z^2}{2}} \quad , \tag{4}$$

and

$$\mathbb{P}\{Z > \mu + z\sigma\} \geq \tfrac{1}{\sqrt{2\pi}}\tfrac{z}{z^2+1}e^{-\frac{z^2}{2}} \quad . \tag{5}$$

## B. Proofs for Lemma 4.1

*Proof of Lemma 4.1.* Let $\mathcal{E}_{i,s}^{\mu}$ denote the event that $|\hat{\mu}_{i,s} - \mu_i| \leq \sqrt{\ln(T)/s}$ holds. Let $\overline{\mathcal{E}_{i,s}^{\mu}}$ denote the complement. We have

$$
\begin{aligned}
\mathbb{P}\left\{\max_{h\in[\phi]}\theta_{i,s}^{(h)} \leq \mu_i\right\} &\leq & \mathbb{P}\{\mathcal{E}_{i,s}^{\mu}\}\,\mathbb{P}\left\{\max_{h\in[\phi]}\theta_{i,s}^{(h)} \leq \mu_i \mid \mathcal{E}_{i,s}^{\mu}\right\} + \mathbb{P}\left\{\overline{\mathcal{E}_{i,s}^{\mu}}\right\} \\
&\leq & \prod_{h\in[\phi]}\mathbb{P}\left\{\theta_{i,s}^{(h)} \leq \hat{\mu}_{i,s} + \sqrt{\ln(T)/s} \mid \mathcal{E}_{i,s}^{\mu}\right\} + 2e^{-2\ln(T)} \\
&= & \prod_{h\in[\phi]}\left(1 - \mathbb{P}\left\{\theta_{1,s}^{(h)} > \hat{\mu}_{1,s} + \sqrt{\ln^{1-\alpha}(T)\ln^{\alpha}(T)/s} \mid \mathcal{E}_{i,s}^{\mu}\right\}\right) + 2/T^2 \\
&\leq^{(a)} & \prod_{h\in[\phi]}\left(1 - \frac{1}{\sqrt{2\pi}}\cdot\frac{\sqrt{\ln^{1-\alpha}(T)}}{\ln^{1-\alpha}(T)+1}e^{-0.5\ln^{1-\alpha}(T)}\right) + 2/T^2 \\
&\leq^{(b)} & \left(1 - \frac{1}{\sqrt{2\pi}}\cdot\frac{\sqrt{\ln^{1-\alpha}(T)}}{\ln^{1-\alpha}(T)+1}e^{-0.5((1-\alpha)\ln(T)+1)}\right)^{\phi} + 2/T^2 \\
&= & \left(1 - \frac{1}{\sqrt{2\pi e}}\cdot\frac{\sqrt{\ln^{1-\alpha}(T)}}{\ln^{1-\alpha}(T)}e^{-0.5((1-\alpha)\ln(T))}\right)^{\phi} + 2/T^2 \\
&\leq^{(c)} & e^{-\phi/\sqrt{2\pi e}\cdot\frac{1}{\sqrt{\ln^{1-\alpha}(T)}}\cdot\frac{1}{T^{0.5(1-\alpha)}}} + 2/T^2 \\
&= & e^{-\sqrt{2\pi e}T^{0.5(1-\alpha)}\ln^{0.5(3-\alpha)}(T)/\sqrt{2\pi e}\cdot\frac{1}{\sqrt{\ln^{1-\alpha}(T)}}\cdot\frac{1}{T^{0.5(1-\alpha)}}} + 2/T^2 \\
&\leq & 3/T,
\end{aligned}
\tag{6}
$$

where step $(a)$ uses the anti-concentration bound shown in (5), step (b) uses $\ln^{1-\alpha}(T) \leq (1-\alpha)\ln(T) + 1$ shown in Fact A.1, and step (c) uses $(1 - x) \leq e^{-x}$. □

# C. Proofs for Lemma C.1

For the case where $\alpha = 0$, Lemma C.1 below is an improved version of Lemma 2.13 in Agrawal & Goyal (2017) and our new results imply both improved problem-dependent and problem-independent regret bounds for Algorithm 2 in Agrawal & Goyal (2017). Assume $T\Delta_i^2 > e$.

**Lemma C.1.** *Let $\theta_{1,s} \sim \mathcal{N}\left(\hat{\mu}_{1,s}, \frac{\ln^\alpha(T)}{s}\right)$. Then, for any integer $s \geq 1$, we have*

$$\mathbb{E}_{\hat{\mu}_{1,s}}\left[\frac{1}{\mathbb{P}\{\theta_{1,s} > \mu_1 | \hat{\mu}_{1,s}\}} - 1\right] \leq 12.34 \quad . \tag{7}$$

*Also, for any integer $s \geq \frac{4(1+\sqrt{2})^2 \ln(T\Delta_i^2)\ln^\alpha(T)}{\Delta_i^2}$, we have*

$$\mathbb{E}_{\hat{\mu}_{1,s}}\left[\frac{1}{\mathbb{P}\{\theta_{1,s} > \mu_1 - \frac{\Delta_i}{2} | \hat{\mu}_{1,s}\}} - 1\right] \leq \frac{72}{T\Delta_i^2} \quad . \tag{8}$$

*Proof.* For the result shown in (7), we analyze two cases: $s = 1$ and $s \geq 2$. For $s = 1$, we have

$$\text{LHS of (7)} = \mathbb{E}\left[\frac{1}{\mathbb{P}\{\theta_{1,s} > \mu_1 | \hat{\mu}_{1,s}\}}\right] - 1 \leq^{(a)} \mathbb{E}\left[\frac{1}{\mathbb{P}\left\{\theta_{1,s} > \hat{\mu}_{1,s} + \sqrt{\frac{\ln^\alpha(T)}{1}} | \hat{\mu}_{1,s}\right\}}\right] - 1 \leq^{(b)} \frac{1}{\frac{1}{\sqrt{2\pi}} \cdot \frac{1}{2} \cdot e^{-0.5}} - 1 \leq 12.176, \tag{9}$$

where step (a) uses $\mu_1 \leq \hat{\mu}_{1,s} + \ln^\alpha(T)$ and step (b) uses the anti-concentration bound shown in (5).

For any $s \geq 2$, since $\hat{\mu}_{1,s}$ is a random variable in $[0, 1]$, we know $|\hat{\mu}_{1,s} - \mu_1| \in [0, 1]$ is also a random variable. Now, we define a sequence of disjoint sub-intervals

$$\left[0, \sqrt{\frac{2\ln(2)}{s}}\right), \left[\sqrt{\frac{2\ln(2)}{s}}, \sqrt{\frac{2\ln(2+1)}{s}}\right), \ldots, \left[\sqrt{\frac{2\ln(r+1)}{s}}, \sqrt{\frac{2\ln(r+2)}{s}}\right), \ldots, \left[\sqrt{\frac{2\ln(r_0(s)+1)}{s}}, \sqrt{\frac{2\ln(r_0(s)+2)}{s}}\right),$$

where $r_0(s)$ is the smallest integer such that $[0, 1] \subseteq \left[0, \sqrt{\frac{2\ln(2)}{s}}\right) \cup \left(\bigcup_{1 \leq r \leq r_0(s)} \left[\sqrt{\frac{2\ln(r+1)}{s}}, \sqrt{\frac{2\ln(r+2)}{s}}\right)\right)$.

We also define events $\mathcal{S}_0 := \left\{|\hat{\mu}_{1,s} - \mu_1| \in \left[0, \sqrt{\frac{2\ln(2)}{s}}\right)\right\}$ and $\mathcal{S}_r := \left\{|\hat{\mu}_{1,s} - \mu_1| \in \left[\sqrt{\frac{2\ln(r+1)}{s}}, \sqrt{\frac{2\ln(r+2)}{s}}\right)\right\}$ for all $1 \leq r \leq r_0(s)$ accordingly.

Now, we have

$$\text{LHS of (7)} = \mathbb{E}\left[\frac{1}{\mathbb{P}\{\theta_{1,s} > \mu_1 | \hat{\mu}_{1,s}\}}\right] - 1 \leq \mathbb{E}\left[\frac{\mathbf{1}\{\mathcal{S}_0\}}{\mathbb{P}\{\theta_{1,s} > \mu_1 | \hat{\mu}_{1,s}\}}\right] + \sum_{1 \leq r \leq r_0(s)} \mathbb{E}\left[\frac{\mathbf{1}\{\mathcal{S}_r\}}{\mathbb{P}\{\theta_{1,s} > \mu_1 | \hat{\mu}_{1,s}\}}\right] - 1 \quad . \tag{10}$$

For the first term in (10), we have

$$\mathbb{E}\left[\frac{\mathbf{1}\{\mathcal{S}_0\}}{\mathbb{P}\{\theta_{1,s} > \mu_1 | \hat{\mu}_{1,s}\}}\right] \leq \mathbb{E}\left[\frac{\mathbf{1}\{\mathcal{S}_0\}}{\mathbb{P}\left\{\theta_{1,s} > \hat{\mu}_{1,s} + \sqrt{\frac{2\ln(2)}{s}} | \hat{\mu}_{1,s}\right\}}\right] \leq \mathbb{E}\left[\frac{\mathbf{1}\{\mathcal{S}_0\}}{\mathbb{P}\left\{\theta_{1,s} > \hat{\mu}_{1,s} + \sqrt{\frac{2\ln(2)\ln^\alpha(T)}{s}} | \hat{\mu}_{1,s}\right\}}\right]$$
$$\leq \frac{1}{\frac{1}{\sqrt{2\pi}} \cdot \frac{\sqrt{2\ln(2)}}{2\ln(2)+1} \cdot e^{-0.5 \cdot 2 \cdot \ln(2)}} \leq 10.161, \tag{11}$$

where the second last inequality uses the anti-concentration bound shown in (5).

For the second term in (10), we have

$$\sum_{1 \leq r \leq r_0(s)} \mathbb{E}\left[\frac{\mathbf{1}\{\mathcal{S}_r\}}{\mathbb{P}\{\theta_{1,s} > \mu_1 | \hat{\mu}_{1,s}\}}\right]$$

$$\leq \sum_{1 \leq r \leq r_0(s)} \mathbb{E}\left[\frac{\mathbf{1}\left\{|\hat{\mu}_{1,s} - \mu_1| \in \left[\sqrt{\frac{2\ln(r+1)}{s}}, \sqrt{\frac{2\ln(r+2)}{s}}\right)\right\}}{\mathbb{P}\{\theta_{1,s} > \mu_1 | \hat{\mu}_{1,s}\}}\right]$$

$$= \sum_{1 \leq r \leq r_0(s)} \mathbb{E}\left[\frac{\mathbf{1}\left\{|\hat{\mu}_{1,s} - \mu_1| \in \left[\sqrt{\frac{2\ln(r+1)}{s}}, \sqrt{\frac{2\ln(r+2)}{s}}\right)\right\}}{\mathbb{P}\left\{\theta_{1,s} > \hat{\mu}_{1,s} + \sqrt{\frac{2\ln(r+2)}{s}} | \hat{\mu}_{1,s}\right\}}\right]$$

$$\leq \sum_{1 \leq r \leq r_0(s)} \mathbb{E}\left[\frac{\mathbf{1}\left\{|\hat{\mu}_{1,s} - \mu_1| \in \left[\sqrt{\frac{2\ln(r+1)}{s}}, \sqrt{\frac{2\ln(r+2)}{s}}\right)\right\}}{\mathbb{P}\left\{\theta_{1,s} > \hat{\mu}_{1,s} + \sqrt{\frac{2\ln(r+2)\ln^\alpha(T)}{s}} | \hat{\mu}_{1,s}\right\}}\right]$$

$$\leq^{(a)} \sum_{1 \leq r \leq r_0(s)} \mathbb{E}\left[\frac{1}{\frac{1}{\sqrt{2\pi}} \cdot \frac{\sqrt{2\ln(r+2)}}{2\ln(r+2)+1} \cdot e^{-0.5 \cdot 2\ln(r+2)}} \cdot \mathbf{1}\left\{|\hat{\mu}_{1,s} - \mu_1| \in \left[\sqrt{\frac{2\ln(r+1)}{s}}, \sqrt{\frac{2\ln(r+2)}{s}}\right)\right\}\right]$$

$$= \sum_{1 \leq r \leq r_0(s)} \frac{\sqrt{2\pi}(2\ln(r+2)+1)}{\sqrt{2\ln(r+2)} \cdot e^{-\ln(r+2)}} \cdot \mathbb{P}\left\{|\hat{\mu}_{1,s} - \mu_1| \geq \sqrt{\frac{2\ln(r+1)}{s}}\right\}$$

$$\leq^{(b)} \sum_{1 \leq r \leq r_0(s)} \frac{\sqrt{2\pi}(2\ln(r+2)+1)}{\sqrt{2\ln(r+2)} \cdot e^{-\ln(r+2)}} \cdot 2e^{-2s \cdot \frac{2\ln(r+1)}{s}}$$

$$= \sum_{1 \leq r \leq r_0(s)} \frac{\sqrt{\pi} \cdot (2\ln(r+2)+1) \cdot (r+2)}{\sqrt{\ln(r+2)}} \cdot 2\frac{1}{(r+1)^4}$$

$$\leq \quad 3.176 \quad ,$$

(12)

where step (a) uses the anti-concentration bound shown in (5) and step (b) uses Hoeffding's inequality.

Plugging the results shown in (11) and (12) into (10), we have

$$\text{LHS of (7)} = \mathbb{E}\left[\frac{1}{\mathbb{P}\{\theta_{1,s} > \mu_1 | \hat{\mu}_{1,s}\}}\right] - 1 \leq \mathbb{E}\left[\frac{\mathbf{1}\{\mathcal{S}_0\}}{\mathbb{P}\{\theta_{1,s} > \mu_1 | \hat{\mu}_{1,s}\}}\right] + \sum_{r \geq 1} \mathbb{E}\left[\frac{\mathbf{1}\{\mathcal{S}_r\}}{\mathbb{P}\{\theta_{1,s} > \mu_1 | \hat{\mu}_{1,s}\}}\right] - 1 \leq 12.34, \quad (13)$$

which concludes the proof of the first result.

For the result shown in (8), we define the following sequence of sub-intervals

$$\left[0, \sqrt{\frac{\ln(T\Delta_i^2)}{s}}\right), \dots, \left[\sqrt{\frac{\ln(r\cdot T\Delta_i^2)}{s}}, \sqrt{\frac{\ln((r+1)\cdot T\Delta_i^2)}{s}}\right), \dots, \left[\sqrt{\frac{\ln(r_0(s)\cdot T\Delta_i^2)}{s}}, \sqrt{\frac{\ln((r_0(s)+1)\cdot T\Delta_i^2)}{s}}\right),$$

where $r_0(s)$ is the smallest integer such that $[0,1] \subseteq \left[0, \sqrt{\frac{\ln(T\Delta_i^2)}{s}}\right) \bigcup_{1 \leq r \leq r_0(s)} \left[\sqrt{\frac{\ln(r\cdot T\Delta_i^2)}{s}}, \sqrt{\frac{\ln((r+1)\cdot T\Delta_i^2)}{s}}\right)$.

We define events $\mathcal{S}_0 := \left\{ |\hat{\mu}_{1,s} - \mu_1| \in \left[0, \sqrt{\frac{\ln(T\Delta_i^2)}{s}}\right) \right\}$ and $\mathcal{S}_r := \left\{ |\hat{\mu}_{1,s} - \mu_1| \in \left[\sqrt{\frac{\ln(rT\Delta_i^2)}{s}}, \sqrt{\frac{\ln((r+1)T\Delta_i^2)}{s}}\right) \right\}$ for all $1 \leq r \leq r_0(s)$ accordingly.

From $s \geq \frac{4(1+\sqrt{2})^2 \ln(T\Delta_i^2)\ln^\alpha(T)}{\Delta_i^2}$, we also have $\Delta_i \geq \sqrt{\frac{4(1+\sqrt{2})^2 \ln(T\Delta_i^2)\ln^\alpha(T)}{s}}$. Then, we have

$$\begin{aligned}
&\text{LHS of (8)}\\
=\ & \mathbb{E}\left[\frac{1}{\mathbb{P}\{\theta_{1,s}>\mu_1-0.5\Delta_i|\hat{\mu}_{1,s}\}}\right] - 1\\[2mm]
\leq\ & \mathbb{E}\left[\frac{1}{\mathbb{P}\left\{\theta_{1,s}>\mu_1-\sqrt{\frac{(1+\sqrt{2})^2\ln(T\Delta_i^2)\ln^\alpha(T)}{s}}\,\middle|\,\hat{\mu}_{1,s}\right\}}\right] - 1\\[2mm]
\leq\ & \left(\mathbb{E}\left[\frac{\mathbf{1}\{\mathcal{S}_0\}}{\mathbb{P}\left\{\theta_{1,s}>\mu_1-\sqrt{\frac{(1+\sqrt{2})^2\ln(T\Delta_i^2)\ln^\alpha(T)}{s}}\,\middle|\,\hat{\mu}_{1,s}\right\}}\right] - 1\right) + \sum_{1\leq r\leq r_0(s)}\mathbb{E}\left[\frac{\mathbf{1}\{\mathcal{S}_r\}}{\mathbb{P}\left\{\theta_{1,s}>\mu_1-\sqrt{\frac{(1+\sqrt{2})^2\ln(T\Delta_i^2)\ln^\alpha(T)}{s}}\,\middle|\,\hat{\mu}_{1,s}\right\}}\right]\\[2mm]
\leq\ & \left(\mathbb{E}\left[\frac{\mathbf{1}\{\mathcal{S}_0\}}{\mathbb{P}\left\{\theta_{1,s}>\mu_1-\sqrt{\frac{(1+\sqrt{2})^2\ln(T\Delta_i^2)\ln^\alpha(T)}{s}}\,\middle|\,\hat{\mu}_{1,s}\right\}}\right] - 1\right) + \sum_{1\leq r\leq r_0(s)}\mathbb{E}\left[\frac{\mathbf{1}\{\mathcal{S}_r\}}{\mathbb{P}\{\theta_{1,s}>\mu_1|\hat{\mu}_{1,s}\}}\right] \quad .
\end{aligned}$$

$$(14)$$

For the first term in (14), we have

$$\begin{aligned}
& \mathbb{E}\left[\frac{\mathbf{1}\{\mathcal{S}_0\}}{\mathbb{P}\left\{\theta_{1,s}>\mu_1-\frac{1}{2}\sqrt{\frac{(1+\sqrt{2})^2\ln(T\Delta_i^2)\ln^\alpha(T)}{s}}\,\middle|\,\hat{\mu}_{1,s}\right\}}\right] - 1\\[2mm]
\leq\ & \mathbb{E}\left[\frac{\mathbf{1}\{\mathcal{S}_0\}}{\mathbb{P}\left\{\theta_{1,s}>\hat{\mu}_{1,s}+\sqrt{\frac{\ln(T\Delta_i^2)}{s}}-\sqrt{\frac{(1+\sqrt{2})^2\ln(T\Delta_i^2)\ln^\alpha(T)}{s}}\,\middle|\,\hat{\mu}_{1,s}\right\}}\right] - 1\\[2mm]
\leq\ & \mathbb{E}\left[\frac{\mathbf{1}\{\mathcal{S}_0\}}{\mathbb{P}\left\{\theta_{1,s}>\hat{\mu}_{1,s}+\sqrt{\frac{\ln(T\Delta_i^2)\ln^\alpha(T)}{s}}-\sqrt{\frac{(1+\sqrt{2})^2\ln(T\Delta_i^2)\ln^\alpha(T)}{s}}\,\middle|\,\hat{\mu}_{1,s}\right\}}\right] - 1\\[2mm]
=\ & \mathbb{E}\left[\frac{\mathbf{1}\{\mathcal{S}_0\}}{\mathbb{P}\left\{\theta_{1,s}>\hat{\mu}_{1,s}-\sqrt{\frac{2\ln(T\Delta_i^2)\ln^\alpha(T)}{s}}\,\middle|\,\hat{\mu}_{1,s}\right\}}\right] - 1\\[2mm]
\overset{(a)}{\leq}\ & \mathbb{E}\left[\frac{1}{1-\frac{0.5}{T\Delta_i^2}}\right] - 1\\[2mm]
\overset{(b)}{\leq}\ & \frac{0.613}{T\Delta_i^2}\ ,
\end{aligned}$$

$$(15)$$

where step (a) uses concentration bound shown in (4) and step (b) uses $\frac{1}{1-\frac{0.5}{T\Delta_i^2}} - 1 = \frac{\frac{0.5}{T\Delta_i^2}}{1-\frac{0.5}{T\Delta_i^2}} \leq \frac{0.5}{T\Delta_i^2}\cdot\frac{1}{1-0.5/e}$.

For the second term in (14), we have

$$
\begin{aligned}
&\sum_{1\le r\le r_0(s)} \mathbb{E}\left[\frac{\mathbf{1}\{\mathcal{S}_r\}}{\mathbb{P}\{\theta_{1,s}>\mu_1|\hat{\mu}_{1,s}\}}\right] \\
&= \sum_{1\le r\le r_0(s)} \mathbb{E}\left[\frac{\mathbf{1}\left\{|\hat{\mu}_{1,s}-\mu_1|\in\left[\sqrt{\frac{\ln(r\cdot T\Delta_i^2)}{s}},\sqrt{\frac{\ln((r+1)\cdot T\Delta_i^2)}{s}}\right)\right\}}{\mathbb{P}\{\theta_{1,s}>\mu_1|\hat{\mu}_{1,s}\}}\right] \\
&\le \sum_{1\le r\le r_0(s)} \mathbb{E}\left[\frac{\mathbf{1}\left\{|\hat{\mu}_{1,s}-\mu_1|\in\left[\sqrt{\frac{\ln(r\cdot T\Delta_i^2)}{s}},\sqrt{\frac{\ln((r+1)\cdot T\Delta_i^2)}{s}}\right)\right\}}{\mathbb{P}\left\{\theta_{1,s}>\hat{\mu}_{1,s}+\sqrt{\frac{\ln((r+1)\cdot T\Delta_i^2)}{s}}|\hat{\mu}_{1,s}\right\}}\right] \\
&\le \sum_{1\le r\le r_0(s)} \mathbb{E}\left[\frac{\mathbf{1}\left\{|\hat{\mu}_{1,s}-\mu_1|\in\left[\sqrt{\frac{\ln(r\cdot T\Delta_i^2)}{s}},\sqrt{\frac{\ln((r+1)\cdot T\Delta_i^2)}{s}}\right)\right\}}{\mathbb{P}\left\{\theta_{1,s}>\hat{\mu}_{1,s}+\sqrt{\frac{\ln((r+1)\cdot T\Delta_i^2)\ln^\alpha(T)}{s}}|\hat{\mu}_{1,s}\right\}}\right] \\
&\le^{(a)} \sum_{1\le r\le r_0(s)} \frac{1}{\frac{1}{2\sqrt{2\pi}}\cdot\frac{1}{\sqrt{\ln((r+1)\cdot T\Delta_i^2)}}\cdot((r+1)\cdot T\Delta_i^2)^{-0.5}}\cdot\mathbb{P}\left\{|\hat{\mu}_{1,s}-\mu_1|\in\left[\sqrt{\frac{\ln(r\cdot T\Delta_i^2)}{s}},\sqrt{\frac{\ln((r+1)\cdot T\Delta_i^2)}{s}}\right)\right\} \\
&\le \sum_{1\le r\le r_0(s)} \frac{1}{\frac{1}{2\sqrt{2\pi}}\cdot\frac{1}{\sqrt{\ln((r+1)\cdot T\Delta_i^2)}}\cdot((r+1)\cdot T\Delta_i^2)^{-0.5}}\cdot\mathbb{P}\left\{|\hat{\mu}_{1,s}-\mu_1|\ge\sqrt{\frac{\ln(r\cdot T\Delta_i^2)}{s}}\right\} \\
&\le^{(b)} \sum_{1\le r\le r_0(s)} \frac{1}{\frac{1}{2\sqrt{2\pi}}\cdot\frac{1}{\sqrt{\ln((r+1)\cdot T\Delta_i^2)}}\cdot((r+1)\cdot T\Delta_i^2)^{-0.5}}\cdot2e^{-2\ln(r\cdot T\Delta_i^2)} \\
&= \sum_{1\le r\le r_0(s)} \frac{1}{\frac{1}{2\sqrt{2\pi}}\cdot\frac{1}{\sqrt{\ln((r+1)\cdot T\Delta_i^2)}}\cdot((r+1)\cdot T\Delta_i^2)^{-0.5}}\cdot2(r\cdot T\Delta_i^2)^{-2} \\
&\le \sum_{1\le r\le r_0(s)} \frac{4\sqrt{2\pi(r+1)\cdot T\Delta_i^2\cdot\ln((r+1)\cdot T\Delta_i^2)}}{(r\cdot T\Delta_i^2)^2} \\
&= \frac{4\sqrt{2\pi}}{T\Delta_i^2}\sum_{1\le r\le r_0(s)}\frac{\sqrt{(r+1)\cdot\ln((r+1)\cdot T\Delta_i^2)}}{r^2\cdot\sqrt{T\Delta_i^2}} \\
&\le \frac{4\sqrt{2\pi}}{T\Delta_i^2}\sum_{1\le r\le r_0(s)}\frac{\sqrt{(r+1)\ln(r+1)}}{r^2}+\frac{\sqrt{r+1}}{r^2} \\
&\le \frac{4\sqrt{2\pi}}{T\Delta_i^2}\times7.034 \\
&\le \frac{70.5235}{T\Delta_i^2}\quad,
\end{aligned}
\tag{16}
$$

where step (a) uses the anti-concentration bound shown in (5), i.e., we have

$$
\begin{aligned}
\mathbb{P}\left\{\theta_{1,s}>\hat{\mu}_{1,s}+\sqrt{\frac{\ln((r+1)\cdot T\Delta_i^2)}{s}}\mid\hat{\mu}_{1,s}\right\} &\ge \mathbb{P}\left\{\theta_{1,s}>\hat{\mu}_{1,s}+\sqrt{\frac{\ln((r+1)\cdot T\Delta_i^2)\ln^\alpha(T)}{s}}\mid\hat{\mu}_{1,s}\right\} \\
&\ge \frac{1}{\sqrt{2\pi}}\cdot\frac{\sqrt{\ln((r+1)\cdot T\Delta_i^2)}}{\ln((r+1)\cdot T\Delta_i^2)+1}\cdot e^{-0.5\cdot\ln((r+1)\cdot T\Delta_i^2)} \\
&= \frac{1}{\sqrt{2\pi}}\cdot\frac{\sqrt{\ln((r+1)\cdot T\Delta_i^2)}}{\ln((r+1)\cdot T\Delta_i^2)+1}\cdot((r+1)\cdot T\Delta_i^2)^{-0.5} \\
&> \frac{1}{\sqrt{2\pi}}\cdot\frac{\sqrt{\ln((r+1)\cdot T\Delta_i^2)}}{2\ln((r+1)\cdot T\Delta_i^2)}\cdot((r+1)\cdot T\Delta_i^2)^{-0.5} \\
&= \frac{1}{2\sqrt{2\pi}}\cdot\frac{1}{\sqrt{\ln((r+1)\cdot T\Delta_i^2)}}\cdot((r+1)\cdot T\Delta_i^2)^{-0.5}\quad.
\end{aligned}
$$

$\square$

## D. Proofs for Theorem 4.2

*Proof.* We first define two high-probability events. For any arm $i \in [K]$, let $\mathcal{E}_i^\mu(t-1) := \left\{ \left| \hat{\mu}_{i,n_i(t-1)} - \mu_i \right| \leq \sqrt{\frac{\ln(T\Delta_i^2)}{n_i(t-1)}} \right\}$ and $\mathcal{E}_i^\theta(t) := \left\{ \theta_i(t) \leq \hat{\mu}_{i,n_i(t-1)} + \sqrt{2\ln(T\Delta_i^2 \cdot \phi)} \cdot \sqrt{\frac{\ln^\alpha(T)}{n_i(t-1)}} \right\}$. Let $\overline{\mathcal{E}_i^\mu(t-1)}$ and $\overline{\mathcal{E}_i^\theta(t)}$ denote the complements, respectively.

Fix a sub-optimal arm $i$. Let $L_i := \frac{(\sqrt{2}+1)^2}{4} \cdot \ln(T\Delta_i^2 \cdot \phi) \cdot \frac{\ln^\alpha(T)}{\Delta_i^2}$ and $r_i^{(*)} = \lceil \log_2(L_i) \rceil$.

Let $t_0$ denote the last round of epoch $r_i^{(*)}$. That is also to say, at the end of round $t_0$, arm $i$'s empirical mean will be updated by using $2^{r_i^{(*)}}$ observations.

Let $N_i(T)$ denote the number of pulls of sub-optimal arm $i$ by the end of round $T$. We upper bound $\mathbb{E}[N_i(T)]$, the expected number of pulls of sub-optimal arm $i$. We decompose the regret based on whether the above-defined events are true or not. We have

$$
\begin{aligned}
\mathbb{E}[N_i(T)] &= \sum_{t=K+1}^T \mathbb{E}[\mathbf{1}\{i_t = i\}] + 1 \\
&= \mathbb{E}\left[\sum_{t=K+1}^{t_0} \mathbf{1}\{i_t = i, n_i(t-1) < L_i\}\right] + \mathbb{E}\left[\sum_{t=t_0+1}^T \mathbf{1}\{i_t = i, n_i(t-1) \geq L_i\}\right] + 1 \\
&\leq \sum_{s=1}^{r_i^{(*)}} 2^s + \sum_{t=K+1}^T \mathbb{E}[\mathbf{1}\{i_t = i, n_i(t-1) \geq L_i\}] + 1 \\
&= \sum_{s=0}^{r_i^{(*)}} 2^s + \sum_{t=K+1}^T \mathbb{E}[\mathbf{1}\{i_t = i, n_i(t-1) \geq L_i\}] \\
&\leq 4L_i + \underbrace{\sum_{t=K+1}^T \mathbb{E}\left[\mathbf{1}\{i_t = i, \mathcal{E}_i^\theta(t), \mathcal{E}_i^\mu(t-1), n_i(t-1) \geq L_i\}\right]}_{\omega_1} \\
&\quad + \underbrace{\sum_{t=K+1}^T \mathbb{E}\left[\mathbf{1}\{i_t = i, \overline{\mathcal{E}_i^\theta(t)}, n_i(t-1) \geq L_i\}\right]}_{\omega_2 = O(1/\Delta_i^2),\ \text{Lemma D.1}} + \underbrace{\sum_{t=K+1}^T \mathbb{E}\left[\mathbf{1}\{i_t = i, \overline{\mathcal{E}_i^\mu(t-1)}, n_i(t-1) \geq L_i\}\right]}_{\omega_3 = O(1/\Delta_i^2),\ \text{Lemma D.2}}.
\end{aligned}
\tag{17}
$$

For $\omega_2$ and $\omega_3$ terms, we prepare a lemma for each of them.

**Lemma D.1.** *We have* $\sum_{t=K+1}^T \mathbb{E}\left[\mathbf{1}\{i_t = i, \overline{\mathcal{E}_i^\theta(t)}, n_i(t-1) \geq L_i\}\right] \leq O\left(\frac{1}{\Delta_i^2}\right)$.

**Lemma D.2.** *We have* $\sum_{t=K+1}^T \mathbb{E}\left[\mathbf{1}\{i_t = i, \overline{\mathcal{E}_i^\mu(t-1)}, n_i(t-1) \geq L_i\}\right] \leq O\left(\frac{1}{\Delta_i^2}\right)$.

The challenging part is to upper bound term $\omega_1$. By tuning $L_i$ properly, we have

$$
\begin{aligned}
\omega_1 &= \sum_{t=K+1}^T \mathbb{E}[\mathbf{1}\{i_t = i, \mathcal{E}_i^\theta(t), \mathcal{E}_i^\mu(t-1), n_i(t-1) \geq L_i\}] \\
&\leq^{(a)} \sum_{t=K+1}^T \mathbb{E}[\mathbf{1}\{i_t = i, \theta_i(t) \leq \mu_i + 0.5\Delta_i\}] \quad,
\end{aligned}
\tag{18}
$$

where step (a) uses the argument that if both events $\mathcal{E}_i^\mu(t-1) = \left\{ \left| \hat{\mu}_{i,n_i(t-1)} - \mu_i \right| \leq \sqrt{\frac{\ln(T\Delta_i^2)}{n_i(t-1)}} \right\}$ and $\mathcal{E}_i^\theta(t) = \left\{ \theta_i(t) \leq \hat{\mu}_{i,n_i(t-1)} + \sqrt{2\ln(T\Delta_i^2 \cdot \phi)} \cdot \sqrt{\frac{\ln^\alpha(T)}{n_i(t-1)}} \right\}$ are true, and $n_i(t-1) \geq L_i$, we have

$$
\begin{aligned}
\theta_i(t) &\leq \hat{\mu}_{i,n_i(t-1)} + \sqrt{2\ln(T\Delta_i^2 \cdot \phi)} \cdot \sqrt{\frac{\ln^\alpha(T)}{n_i(t-1)}} \\
&\leq \mu_i + \sqrt{\frac{\ln(T\Delta_i^2)}{n_i(t-1)}} + \sqrt{2\ln(T\Delta_i^2 \cdot \phi)} \cdot \sqrt{\frac{\ln^\alpha(T)}{n_i(t-1)}} \\
&< \mu_i + \sqrt{\frac{\ln(T\Delta_i^2 \cdot \phi)}{L_i}} \sqrt{\ln^\alpha(T)} + \sqrt{2\ln(T\Delta_i^2 \cdot \phi)} \cdot \sqrt{\frac{\ln^\alpha(T)}{L_i}} \\
&< \mu_i + (\sqrt{2} + 1)\sqrt{\frac{\ln(T\Delta_i^2 \cdot \phi)}{L_i}} \sqrt{\ln^\alpha(T)} \\
&= \mu_i + 0.5\Delta_i \quad,
\end{aligned}
\tag{19}
$$

where the last step applies $L_i = \frac{(\sqrt{2}+1)^2}{4} \cdot \ln(T\Delta_i^2 \cdot \phi) \cdot \frac{\ln^\alpha(T)}{\Delta_i^2}$.

Since the optimal arm 1 can either be in the mandatory TS-Gaussian phase or the optional UCB phase, we continue decomposing the regret based on the case of the optimal arm 1. Define $\mathcal{T}_1(t)$ as the event that the optimal arm 1 in round $t$ is in the mandatory TS-Gaussian phase, that is, using a fresh Gaussian mean reward model in the learning. Let $\overline{\mathcal{T}_1(t)}$ denote the complement, that is, using $\text{MAX}_1 = \max_{h_1 \in [\phi]} \theta_{1,n_1(t-1)}^{(h_1)}$ in the learning, where all $\theta_{1,n_1(t-1)}^{(h_1)} \sim \mathcal{N}\left(\hat{\mu}_{1,n_1(t-1)}, \frac{\ln^\alpha(T)}{n_1(t-1)}\right)$ are i.i.d. random variables.

We have

$$\omega_1 \leq \underbrace{\sum_{t=K+1}^{T} \mathbb{E}\left[\mathbf{1}\left\{i_t = i, \theta_i(t) \leq \mu_i + 0.5\Delta_i, \mathcal{T}_1(t)\right\}\right]}_{I_1} + \underbrace{\sum_{t=K+1}^{T} \mathbb{E}\left[\mathbf{1}\left\{i_t = i, \theta_i(t) \leq \mu_i + 0.5\Delta_i, \overline{\mathcal{T}_1(t)}\right\}\right]}_{I_2}. \quad (20)$$

**Upper bound $I_1$.** Note that if event $\mathcal{T}_1(t)$ is true, we know the optimal arm 1 is using a fresh Gaussian mean reward model in the learning in round $t$, that is, $\theta_1(t) \sim \mathcal{N}\left(\hat{\mu}_{1,n_1(t-1)}, \frac{\ln^\alpha(T)}{n_1(t-1)}\right)$. Term $I_1$ will use a similar analysis to Lemma 2.8 of Agrawal & Goyal (2017), which links the probability of pulling a sub-optimal arm $i$ to the probability of pulling the optimal arm 1. We formalize this into our technical Lemma D.3 below. Let $\mathcal{F}_{t-1} = \{h_1(\tau), h_2(\tau), \ldots, h_K(\tau), i_\tau, X_{i_\tau}(\tau), \forall \tau = 1, 2, \ldots, t-1\}$ collect all the history information by the end of round $t-1$. It collects the number of unused Gaussian sampling budget $h_i(\tau)$ by the end of round $\tau$ for all $i \in [K]$, the index $i_\tau$ of the pulled arm, and the observed reward $X_{i_\tau}(\tau)$ for all rounds $\tau = 1, 2, \ldots, t-1$. Let $\theta_{1,n_1(t-1)} \sim \mathcal{N}\left(\hat{\mu}_{1,n_1(t-1)}, \frac{\ln^\alpha(T)}{n_1(t-1)}\right)$ be a Gaussian random variable.

**Lemma D.3.** *For any instantiation $F_{t-1}$ of $\mathcal{F}_{t-1}$, we have*

$$\mathbb{E}\left[\mathbf{1}\left\{i_t = i, \mathcal{T}_1(t), \theta_i(t) \leq \mu_i + 0.5\Delta_i\right\} \mid \mathcal{F}_{t-1} = F_{t-1}\right]$$
$$\leq \left(\frac{1}{\mathbb{P}\left\{\theta_{1,n_1(t-1)} > \mu_1 - 0.5\Delta_i \mid \mathcal{F}_{t-1} = F_{t-1}\right\}} - 1\right) \mathbb{E}\left[\mathbf{1}\left\{i_t = 1\right\} \mid \mathcal{F}_{t-1} = F_{t-1}\right]. \quad (21)$$

With Lemma D.3 in hand, we upper bound term $I_1$. Let $L_{1,i} := \frac{4(1+\sqrt{2})^2 \ln(T\Delta_i^2) \ln^\alpha(T)}{\Delta_i^2}$. Let $r_1^{(*)} = \lceil \log_2(L_{1,i}) \rceil$. We have

$$
\begin{aligned}
I_1 &= \sum_{t=K+1}^{T} \mathbb{E}\left[\mathbf{1}\left\{i_t = i, \theta_i(t) \leq \mu_i + 0.5\Delta_i, \mathcal{T}_1(t)\right\}\right] \\
&= \sum_{t=K+1}^{T} \mathbb{E}\left[\mathbb{E}\left[\mathbf{1}\left\{i_t = i, \theta_i(t) \leq \mu_i + 0.5\Delta_i, \mathcal{T}_1(t)\right\} \mid \mathcal{F}_{t-1}\right]\right] \\
&\leq \sum_{t=K+1}^{T} \mathbb{E}\left[\left(\frac{1}{\mathbb{P}\left\{\theta_{1,n_1(t-1)} > \mu_1 - 0.5\Delta_i \mid \mathcal{F}_{t-1} = F_{t-1}\right\}} - 1\right) \cdot \mathbb{E}\left[\mathbf{1}\left\{i_t = 1\right\} \mid \mathcal{F}_{t-1} = F_{t-1}\right]\right] \\
&= \sum_{t=K+1}^{T} \mathbb{E}\left[\mathbb{E}\left[\left(\frac{1}{\mathbb{P}\left\{\theta_{1,n_1(t-1)} > \mu_1 - 0.5\Delta_i \mid \mathcal{F}_{t-1} = F_{t-1}\right\}} - 1\right) \cdot \mathbf{1}\left\{i_t = 1\right\} \mid \mathcal{F}_{t-1} = F_{t-1}\right]\right] \\
&= \sum_{t=K+1}^{T} \mathbb{E}\left[\left(\frac{1}{\mathbb{P}\left\{\theta_{1,n_1(t-1)} > \mu_1 - 0.5\Delta_i \mid \mathcal{F}_{t-1} = F_{t-1}\right\}} - 1\right) \cdot \mathbf{1}\left\{i_t = 1\right\}\right] \\
&\leq \sum_{s=0}^{\log(T)} 2^{s+1} \cdot \mathbb{E}\left[\left(\frac{1}{\mathbb{P}\left\{\theta_{1,2^s} > \mu_1 - 0.5\Delta_i \mid \hat{\mu}_{1,2^s}\right\}} - 1\right)\right] \\
&\leq \underbrace{\sum_{s=0}^{r_1^{(*)}-1} 2^{s+1} \cdot \mathbb{E}\left[\left(\frac{1}{\mathbb{P}\left\{\theta_{1,2^s} > \mu_1 - 0.5\Delta_i \mid \hat{\mu}_{1,2^s}\right\}} - 1\right)\right]}_{\leq 12.34 \text{ from } (7)} + \underbrace{\sum_{s=r_1^{(*)}}^{\log(T)} 2^{s+1} \cdot \mathbb{E}\left[\left(\frac{1}{\mathbb{P}\left\{\theta_{1,2^s} > \mu_1 - 0.5\Delta_i \mid \hat{\mu}_{1,2^s}\right\}} - 1\right)\right]}_{\leq \frac{72}{T\Delta_i^2} \text{ from } (8)} \\
&\leq 4 \cdot L_{1,i} \cdot 12.34 + \sum_{s=r_1^{(*)}}^{\log(T)} 2^{s+1} \cdot \frac{72}{T\Delta_i^2} \\
&\leq 50 L_{1,i} + O(1/\Delta_i^2) .
\end{aligned}
$$

$$(22)$$

**Upper bound $I_2$.** Note that if event $\mathcal{T}_1(t)$ is false, we know the optimal arm 1 is using $\text{MAX}_1 = \max_{h_1 \in [\phi]} \theta_{1,n_1(t-1)}^{(h_1)}$ in the learning, where $\theta_{1,n_1(t-1)}^{(h_1)} \sim \mathcal{N}\left(\hat{\mu}_{1,n_1(t-1)}, \frac{\ln^\alpha(T)}{n_1(t-1)}\right)$ for each $h_1 \in [\phi]$. We have

$$
\begin{aligned}
I_2 &= \sum_{t=K+1}^{T} \mathbb{E}\left[\mathbf{1}\left\{i_t = i, \theta_i(t) \le \mu_i + 0.5\Delta_i, \overline{\mathcal{T}_1(t)}\right\}\right] \\
&< \sum_{t=K+1}^{T} \mathbb{E}\left[\mathbf{1}\left\{i_t = i, \theta_i(t) \le \mu_i + \Delta_i, \overline{\mathcal{T}_1(t)}\right\}\right] \\
&\le \sum_{t=K+1}^{T} \mathbb{E}\left[\mathbf{1}\left\{i_t = i, \theta_1(t) \le \mu_i + \Delta_i, \overline{\mathcal{T}_1(t)}\right\}\right] \\
&\le \sum_{t=K+1}^{T} \sum_{s=0}^{\log(T)} \underbrace{\mathbb{E}\left[\mathbf{1}\left\{\max_{h_1 \in [\phi]} \theta_{1,2^s}^{(h_1)} \le \mu_1\right\}\right]}_{\text{Lemma 4.1}} \\
&\le \sum_{t=K+1}^{T} \sum_{s=0}^{\log(T)} O(1/T) \\
&\le O(\ln(T)) \quad .
\end{aligned}
$$
(23)

From (22) and (23), we have $\omega_1 \le I_1 + I_2 \le 50L_{1,i} + O(1/\Delta_i^2) + O(\ln(T)) \le O\left(\frac{\ln(T\Delta_i^2)\ln^\alpha(T)}{\Delta_i^2}\right)$, which gives

$$
\mathbb{E}[N_i(T)] \le O\left(\frac{\ln(\phi T \Delta_i^2)\ln^\alpha(T)}{\Delta_i^2}\right) + O\left(\frac{\ln(T\Delta_i^2)\ln^\alpha(T)}{\Delta_i^2}\right) = O\left(\frac{\ln(\phi T \Delta_i^2)\ln^\alpha(T)}{\Delta_i^2}\right) \quad .
$$
(24)

Therefore, the problem-dependent regret bound by the end of round $T$ is

$$
\begin{aligned}
&\sum_{i\in[K]:\Delta_i>0} \mathbb{E}[N_i(T)] \cdot \Delta_i \\
=\ & \sum_{i\in[K]:\Delta_i>0} O\left(\frac{\ln(\phi T \Delta_i^2)\ln^\alpha(T)}{\Delta_i}\right) \\
=\ & \sum_{i\in[K]:\Delta_i>0} O\left(\frac{\ln(c_0 T^{0.5(1-\alpha)}\ln^{0.5(3-\alpha)}(T)T\Delta_i^2)\ln^\alpha(T)}{\Delta_i}\right) \\
\le\ & \sum_{i\in[K]:\Delta_i>0} O\left(\frac{\ln(T^{0.5(3-\alpha)}\Delta_i^2)\ln^\alpha(T)}{\Delta_i}\right) + O\left(\frac{(3-\alpha)\ln\ln(T)\ln^\alpha(T)}{\Delta_i}\right) \quad .
\end{aligned}
$$
(25)

For the proof of worst-case regret bound, we set the critical gap $\Delta_* := \sqrt{K\ln^{1+\alpha}(T)/T}$. The regret from pulling any sub-optimal arms with mean reward gaps no greater than $\Delta_*$ is at most $T\Delta_* = O(\sqrt{KT\ln^{1+\alpha}(T)})$. The regret from pulling any sub-optimal arms with mean reward gaps greater than $\Delta_*$ is at most $\sum_{i\in[K]:\Delta_i\ge\Delta_*} O\left(\frac{\ln(T^{0.5(3-\alpha)}\Delta_i^2)\ln^\alpha(T)}{\Delta_i}\right) +$

$O\left(\frac{(3-\alpha)\ln\ln(T)\ln^\alpha(T)}{\Delta_i}\right) \le \sum_{i\in[K]:\Delta_i\ge\Delta_*} O\left(\frac{\ln(T)\ln^\alpha(T)}{\Delta_*}\right) + O\left(\frac{\ln\ln(T)\ln^\alpha(T)}{\Delta_*}\right) \le O\left(\sqrt{KT\ln^{1+\alpha}(T)}\right).$ $\qquad\square$

*Proof of Lemma D.1.* Let $\tau_s^{(i)}$ be the round by the end of which the empirical mean will be computed based on $2^s$ fresh observations.

We have

$$
\begin{aligned}
& \sum_{t=K+1}^{T} \mathbb{E}\left[\mathbf{1}\left\{i_t = i, \overline{\mathcal{E}_i^\theta(t)}, n_i(t-1) \geq L_i\right\}\right] \\
= & \sum_{t=K+1}^{T} \mathbb{E}\left[\mathbf{1}\left\{i_t = i, \theta_i(t) > \hat{\mu}_{i,n_i(t-1)} + \sqrt{2\ln(T\Delta_i^2 \cdot \phi)} \cdot \sqrt{\frac{\ln^\alpha(T)}{n_i(t-1)}}, n_i(t-1) \geq L_i\right\}\right] \\
\leq & \sum_{s=0}^{\log(T)} \mathbb{E}\left[\sum_{t=\tau_s^{(i)}+1}^{\tau_{s+1}^{(i)}} \mathbf{1}\left\{i_t = i, \theta_i(t) > \hat{\mu}_{i,n_i(t-1)} + \sqrt{2\ln(T\Delta_i^2 \cdot \phi)} \cdot \sqrt{\frac{\ln^\alpha(T)}{n_i(t-1)}}\right\}\right] \\
\leq & \sum_{s=0}^{\log(T)} 2^{s+1} \cdot \mathbb{P}\left\{\mathrm{MAX}_i > \hat{\mu}_{i,2^s} + \sqrt{2\ln(T\Delta_i^2 \cdot \phi)} \cdot \sqrt{\frac{\ln^\alpha(T)}{2^s}}\right\} \\
\leq & \sum_{s=0}^{\log(T)} 2^{s+1} \cdot \phi \cdot \frac{1}{2} e^{-\ln(T\Delta_i^2 \cdot \phi)} \\
\leq & O\left(T \cdot \phi \cdot \frac{1}{T\Delta_i^2 \cdot \phi}\right) \\
\leq & O\left(\frac{1}{\Delta_i^2}\right) \quad ,
\end{aligned}
\tag{26}
$$

which concludes the proof.

$\square$

*Proof of Lemma D.2.* From Hoeffding's inequality, we have

$$
\begin{aligned}
& \sum_{t=K+1}^{T} \mathbb{E}\left[\mathbf{1}\left\{i_t = i, \overline{\mathcal{E}_i^\mu(t-1)}, n_i(t-1) \geq L_i\right\}\right] \\
\leq & \sum_{s=0}^{\log(T)} \mathbb{P}\left\{|\hat{\mu}_{i,2^s} - \mu_i| \leq \sqrt{\frac{\ln(T\Delta_i^2)}{2^s}}\right\} 2^{s+1} \\
\leq & \sum_{s=0}^{\log(T)} 2e^{-2\ln(T\Delta_i^2)} \cdot 2^{s+1} \\
\leq & O\left(T \cdot \frac{1}{T\Delta_i^2 \cdot T\Delta_i^2}\right) \\
\leq & O\left(\frac{1}{\Delta_i^2}\right) \quad ,
\end{aligned}
\tag{27}
$$

which concludes the proof. $\square$

*Proof of Lemma D.3.* For any $F_{t-1}$, we have

$$
\begin{aligned}
& \mathbb{E}\left[\mathbf{1}\left\{i_t = i, \theta_i(t) \leq \mu_i + 0.5\Delta_i, \mathcal{T}_1(t)\right\} \mid \mathcal{F}_{t-1} = F_{t-1}\right] \\
\leq & \mathbf{1}\left\{\mathcal{T}_1(t)\right\} \cdot \mathbb{E}\left[\mathbf{1}\left\{\theta_1(t) \leq \mu_i + 0.5\Delta_i, \theta_j(t) \leq \mu_i + 0.5\Delta_i, \forall j \in [K] \setminus \{1\}\right\} \mid \mathcal{F}_{t-1} = F_{t-1}\right] \\
= & \mathbf{1}\left\{\mathcal{T}_1(t)\right\} \cdot \mathbb{E}\left[\mathbf{1}\left\{\theta_1(t) \leq \mu_i + 0.5\Delta_i\right\} \mid \mathcal{F}_{t-1} = F_{t-1}\right] \cdot \mathbb{E}\left[\mathbf{1}\left\{\theta_j(t) \leq \mu_i + 0.5\Delta_i, \forall j \in [K] \setminus \{1\}\right\} \mid \mathcal{F}_{t-1} = F_{t-1}\right],
\end{aligned}
\tag{28}
$$

where the first inequality uses the fact that event $\mathcal{T}_1(t)$ is determined by the history information. Note if $h_1(t-1) \in [\phi]$, we have $\mathbf{1}\left\{\mathcal{T}_1(t)\right\} = 1$; if $h_1(t-1) = 0$, we have $\mathbf{1}\left\{\mathcal{T}_1(t)\right\} = 0$.

We also have

$$
\begin{aligned}
& \mathbb{E}\left[\mathbf{1}\left\{i_t = 1, \theta_i(t) \leq \mu_i + 0.5\Delta_i, \mathcal{T}_1(t)\right\} \mid \mathcal{F}_{t-1} = F_{t-1}\right] \\
\geq & \mathbf{1}\left\{\mathcal{T}_1(t)\right\} \cdot \mathbb{E}\left[\mathbf{1}\left\{\theta_1(t) > \mu_i + 0.5\Delta_i \geq \theta_j(t), \forall j \in [K] \setminus \{1\}\right\} \mid \mathcal{F}_{t-1} = F_{t-1}\right] \\
= & \mathbf{1}\left\{\mathcal{T}_1(t)\right\} \cdot \mathbb{E}\left[\mathbf{1}\left\{\theta_1(t) > \mu_i + 0.5\Delta_i\right\} \mid \mathcal{F}_{t-1} = F_{t-1}\right] \cdot \mathbb{E}\left[\mathbf{1}\left\{\theta_j(t) \leq \mu_i + 0.5\Delta_i, \forall j \in [K] \setminus \{1\}\right\} \mid \mathcal{F}_{t-1} = F_{t-1}\right].
\end{aligned}
\tag{29}
$$

Now, we categorize all the possible $F_{t-1}$'s of $\mathcal{F}_{t-1}$ into two groups based on whether $\mathbf{1}\left\{\mathcal{T}_1(t)\right\} = 0$ or $\mathbf{1}\left\{\mathcal{T}_1(t)\right\} = 1$.

**Case 1:** For any $F_{t-1}$ such that $\mathbf{1}\{\mathcal{T}_1(t)\} = 0$, combining (28) and (29) gives

$$
\begin{aligned}
&\mathbb{E}\left[\mathbf{1}\left\{i_t = i, \theta_i(t) \le \mu_i + 0.5\Delta_i, \mathcal{T}_1(t)\right\} \mid \mathcal{F}_{t-1} = F_{t-1}\right] \\
=\ & 0 \\
\le\ & \left(\frac{1}{\mathbb{P}\left\{\theta_{1,n_1(t-1)} > \mu_1 + 0.5\Delta_i \mid \mathcal{F}_{t-1} = F_{t-1}\right\}} - 1\right) \cdot \mathbb{E}\left[\mathbf{1}\{i_t = 1\} \mid \mathcal{F}_{t-1} = F_{t-1}\right] \quad,
\end{aligned}
\tag{30}
$$

where the last equality uses the fact that $0 < \left(\frac{1}{\mathbb{P}\left\{\theta_{1,n_1(t-1)} > \mu_i + 0.5\Delta_i \mid \mathcal{F}_{t-1} = F_{t-1}\right\}} - 1\right) < +\infty.$

**Case 2:** For any $F_{t-1}$ such that $\mathbf{1}\{\mathcal{T}_1(t)\} = 1$, we have

$$
\begin{aligned}
&\mathbb{E}\left[\mathbf{1}\left\{i_t = i, \theta_i(t) \le \mu_i + 0.5\Delta_i, \mathcal{T}_1(t)\right\} \mid \mathcal{F}_{t-1} = F_{t-1}\right] \\
\le\ & \mathbf{1}\{\mathcal{T}_1(t)\} \cdot \mathbb{E}\left[\mathbf{1}\left\{\theta_1(t) \le \mu_i + 0.5\Delta_i, \theta_j(t) \le \mu_i + 0.5\Delta_i, \forall j \in [K] \setminus \{1\}\right\} \mid \mathcal{F}_{t-1} = F_{t-1}\right] \\
=\ & \mathbf{1}\{\mathcal{T}_1(t)\} \cdot \mathbb{E}\left[\mathbf{1}\left\{\theta_1(t) \le \mu_i + 0.5\Delta_i\right\} \mid \mathcal{F}_{t-1} = F_{t-1}\right] \cdot \mathbb{E}\left[\mathbf{1}\left\{\theta_j(t) \le \mu_i + 0.5\Delta_i, \forall j \in [K] \setminus \{1\}\right\} \mid \mathcal{F}_{t-1} = F_{t-1}\right] \\
=\ & \mathbb{E}\left[\mathbf{1}\left\{\theta_{1,n_1(t-1)} \le \mu_i + 0.5\Delta_i\right\} \mid \mathcal{F}_{t-1} = F_{t-1}\right] \cdot \mathbb{E}\left[\mathbf{1}\left\{\theta_j(t) \le \mu_i + 0.5\Delta_i, \forall j \in [K] \setminus \{1\}\right\} \mid \mathcal{F}_{t-1} = F_{t-1}\right] \quad.
\end{aligned}
\tag{31}
$$

We also have

$$
\begin{aligned}
&\mathbb{E}\left[\mathbf{1}\left\{i_t = 1, \theta_i(t) \le \mu_i + 0.5\Delta_i, \mathcal{T}_1(t)\right\} \mid \mathcal{F}_{t-1} = F_{t-1}\right] \\
\ge\ & \mathbf{1}\{\mathcal{T}_1(t)\} \cdot \mathbb{E}\left[\mathbf{1}\left\{\theta_1(t) > \mu_i + 0.5\Delta_i \ge \theta_j(t), \forall j \in [K] \setminus \{1\}\right\} \mid \mathcal{F}_{t-1} = F_{t-1}\right] \\
=\ & \mathbf{1}\{\mathcal{T}_1(t)\} \cdot \mathbb{E}\left[\mathbf{1}\left\{\theta_1(t) > \mu_i + 0.5\Delta_i\right\} \mid \mathcal{F}_{t-1} = F_{t-1}\right] \cdot \mathbb{E}\left[\mathbf{1}\left\{\theta_j(t) \le \mu_i + 0.5\Delta_i, \forall j \in [K] \setminus \{1\}\right\} \mid \mathcal{F}_{t-1} = F_{t-1}\right] \\
=\ & \underbrace{\mathbb{E}\left[\mathbf{1}\left\{\theta_{1,n_1(t-1)} > \mu_i + 0.5\Delta_i\right\} \mid \mathcal{F}_{t-1} = F_{t-1}\right]}_{>0} \cdot \mathbb{E}\left[\mathbf{1}\left\{\theta_j(t) \le \mu_i + 0.5\Delta_i, \forall j \in [K] \setminus \{1\}\right\} \mid \mathcal{F}_{t-1} = F_{t-1}\right] \quad.
\end{aligned}
\tag{32}
$$

From (31) and (32), we have

$$
\begin{aligned}
&\mathbb{E}\left[\mathbf{1}\left\{i_t = i, \theta_i(t) \le \mu_i + 0.5\Delta_i, \mathcal{T}_1(t)\right\} \mid \mathcal{F}_{t-1} = F_{t-1}\right] \\
\le\ & \frac{\mathbb{P}\left\{\theta_{1,n_1(t-1)} \le \mu_i + 0.5\Delta_i \mid \mathcal{F}_{t-1} = F_{t-1}\right\}}{\mathbb{P}\left\{\theta_{1,n_1(t-1)} > \mu_i + 0.5\Delta_i \mid \mathcal{F}_{t-1} = F_{t-1}\right\}} \cdot \mathbb{E}\left[\mathbf{1}\left\{i_t = 1, \theta_i(t) \le \mu_i + 0.5\Delta_i, \mathcal{T}_1(t)\right\} \mid \mathcal{F}_{t-1} = F_{t-1}\right] \\
\le\ & \left(\frac{1}{\mathbb{P}\left\{\theta_{1,n_1(t-1)} > \mu_i + 0.5\Delta_i \mid \mathcal{F}_{t-1} = F_{t-1}\right\}} - 1\right) \cdot \mathbb{E}\left[\mathbf{1}\{i_t = 1\} \mid \mathcal{F}_{t-1} = F_{t-1}\right] \quad,
\end{aligned}
\tag{33}
$$

which concludes the proof. $\qquad\square$

# E. Additional Experimental Results

### E.1. M-TS-Gaussian parameter selection

Recall that in Section 5.3, we let $c = \frac{1}{2c_0(b+1)} T^{0.5(1+\alpha)} \ln^{-1.5(1-\alpha)}(T)$ for any $b$ for M-TS-Gaussian to satisfy $\sqrt{2c_0 T^{0.5(1-\alpha)} \ln^{1.5(1-\alpha)}(T)}$-GDP. To determine the best $b$ value for each $\alpha$ considered in Section 5.3, we conduct experiments with $b = \{0, 1, 500, 1000, 2000, 5000, 100000\}$. The results are shown in Figure 6.

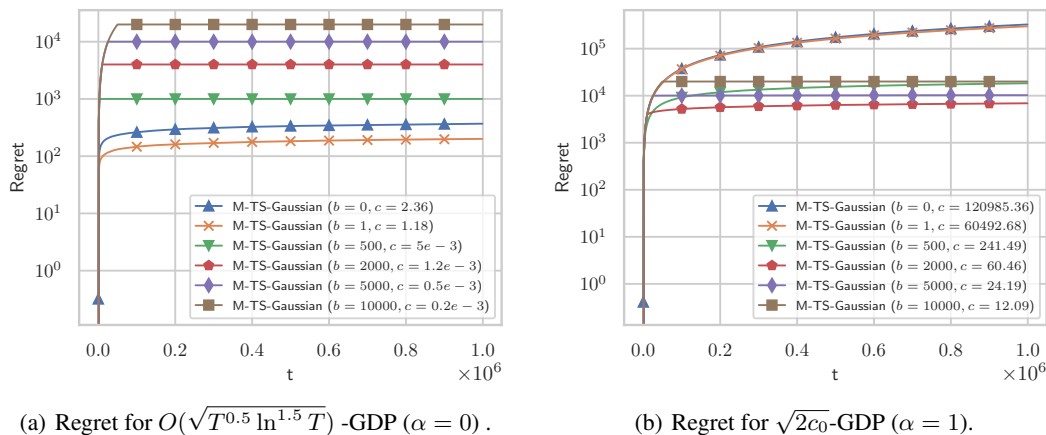

(a) Regret for $O(\sqrt{T^{0.5} \ln^{1.5} T})$ -GDP ($\alpha = 0$).

(b) Regret for $\sqrt{2c_0}$-GDP ($\alpha = 1$).

*Figure 6.* The regret of M-TS-Gaussian with parameters $b = \{0, 1, 500, 1000, 2000, 5000, 100000\}$ under two privacy guarantees.

We can observe that when $\alpha = 0$, M-TS-Gaussian achieves the lowest regret with $b = 1$ and $c = 1.18$, as shown in Figure 6(a). When $\alpha = 1$, M-TS-Gaussian achieves the lowest regret with $b = 2000$ and $c = 60.46$, as shown in Figure 6(b).

### E.2. Comparison with $(\epsilon, 0)$-DP algorithms

We compare DP-TS-UCB with $(\varepsilon, 0)$-DP algorithms with $\varepsilon = 0.5$: DP-SE (Sajed & Sheffet, 2019), Anytime-Lazy-UCB (Hu et al., 2021) and Lazy-DP-TS (Hu & Hegde, 2022). These algorithms use the Laplace mechanism to inject noise.

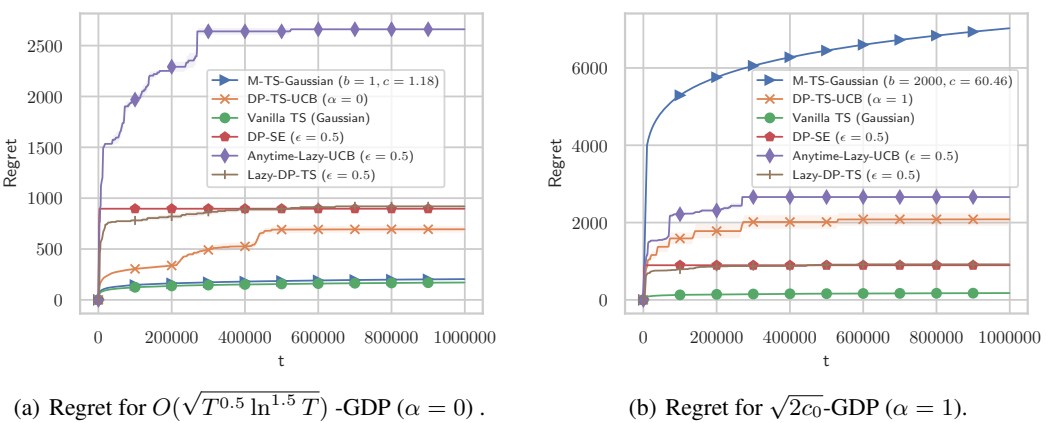

(a) Regret for $O(\sqrt{T^{0.5} \ln^{1.5} T})$ -GDP ($\alpha = 0$).

(b) Regret for $\sqrt{2c_0}$-GDP ($\alpha = 1$).

*Figure 7.* The regret of DP-TS-UCB and M-TS-Gaussian under the same privacy guarantee with $\alpha = 0$ and 1, with comparison to $(\epsilon, 0)$-DP algorithms.

We can see that when $\alpha = 0$, both DP-TS-UCB and M-TS-Gaussian perform better than the $(\epsilon, 0)$-DP algorithms, as shown in Figure 7(a). When we increase $\alpha = 1$, M-TS-Gaussian performs worse than the $(\epsilon, 0)$-DP algorithms, but DP-TS-UCB still outperforms Anytime-Lazy-UCB, as shown in Figure 7(b).

