# OpenReview forum: "Connecting Thompson Sampling and UCB: Towards More Efficient Trade-offs Between Privacy and Regret"
_ICML.cc/2025/Conference — ICML 2025 poster_

### Official Review · Reviewer_95KC · 2025-03-09

**Overall Recommendation:** 3

**Summary:**

This paper studies the problem of differentially private stochastic bandit. It proposes a new algorithm which is roughly Thompson sampling, with an option to re-use previous samples. Their algorithm offers a trade-off between regret and privacy, a strict improvement over previous works on the privacy when maintaining a near-optimal regret.

## update after rebuttal

I have no other concerns. Please consider (1) justifying or modifying the "UCB" naming, and (2) adding a discussion on the trade-off of $\alpha$ in a future version.

**Claims And Evidence:**

It's unclear to me how much similarity there is between UCB and the algorithm in this paper. According to line 10 of the algorithm, the re-using phase where we set $\theta_i(t)$ as the maximum of previous samples is considered the only UCB-style part. The justification, Lemma 4.1, seems to be a bound similarity instead of algorithmic similarity. I'm not sure if calling such mechanism UCB is misleading, since there is no upper confidence bound involved.

**Essential References Not Discussed:**

"TS-UCB: Improving on Thompson Sampling With Little to No Additional Computation", by Jackie Baek and Vivek F. Farias, AISTATS2023. Although this paper doesn't study DP, I would appreciate a discussion on algorithmic and idea similarities/differences between the two works.

**Experimental Designs Or Analyses:**

Yes.

**Methods And Evaluation Criteria:**

Yes.

**Other Comments Or Suggestions:**

No.

**Other Strengths And Weaknesses:**

The new technique of re-using previous samples is smart and interesting, which plays a crucial role in the improvement over previous works.

In the title, it should be "Thompson" instead of "Thomson".

In table 1, the use of $\log$ and $\ln$ is inconsistent.

**Questions For Authors:**

I wonder if the trade-off is necessary, when constant privacy is achievable at the cost of only a logarithmic term in regret. Does an $O(\sqrt{\ln T})$ term matter a lot in regret? In my opinion it doesn't, and it seems setting $\alpha=1$ is a strictly better choice.

**Relation To Broader Scientific Literature:**

This work is of potential interest to the privacy community, besides the ML community.

**Theoretical Claims:**

I only checked the proof sketch of Theorem 4.2 and the proof of Theorem 4.4. The proof sketch of Theorem 4.2 is intuitive, but I would like to see more details on the step of putting everything together. The proof of Theorem 4.4 is too casually written.

---

> ### Author Rebuttal · Authors · 2025-04-01
>
> Thank you very much for the constructive comments. We address each of your questions as follows.
>
> (1) Regarding the **similarity between our proposed algorithm and UCB**,
> as theoretically justified in Lemma 4.1 and the content just above it (Lines 266 to 271), the reason why we call it UCB is that the maximum value of previous samples behaves like the upper confidence bound in the classical UCB1 algorithm.  We agree with the comment that the UCB part in our proposed algorithm is a kind of bound similarity as our proposed algorithm itself does not explicitly construct upper confidence bounds. We appreciate your concern about the potential misinterpretation and will refine our writing to ensure clarity.
>
> (2) Regarding the proof of our presented theorems, we appreciate your suggestions. For Theorem 4.2, we will clarify more key steps by bringing some of the details currently in Appendix D into the main text to provide a more complete picture of the proof sketch. For Theorem 4.4, we will complement the existing proof with a more formal, math-style proof to enhance its rigor and clarity in the appendix.
>
> (3) Regarding **"TS-UCB: Improving on Thompson Sampling With Little to No Additional Computation"**, by Jackie Baek and Vivek F. Farias, AISTATS2023, we thank you very much for referring us to this relevant paper.
>
> After carefully reviewing it, in addition to our focus on privacy, we identify two other key differences between their TS-UCB and our proposed algorithm. (1) Their work focuses on Bayesian regret, whereas our analysis is in the frequentist setting. Since Bayesian regret bound cannot be translated to frequentist regret bound easily, this represents a fundamental distinction. (2) While their TS-UCB samples multiple posterior models, it aggregates them by averaging. The averaging will not behave like an upper confidence bound (as opposed to our maximum), which does not provide sufficient exploration. We will incorporate these points into our discussion of related work.
>
>
> (4) The reasons  **why we provide  tuning $\alpha \in [0,1]$ to trade off privacy and regret**:
>
> From a theoretical standpoint, it provides insights into the interplay between the privacy and regret while also offering the privacy-regret trade-off flexibility by tuning the parameter $\alpha$ to balance these two. This perspective helps us better understand the fundamental limits of private decision-making. For the stochastic bandit community, we care more about the regret bounds, and, thus we would like to avoid the extra $\ln T$ factor. From the perspective of privacy, we agree that in practical scenarios, setting $\alpha = 1$ may often be the preferable choice, as the extra $\ln T$ factor is typically negligible.  Achieving a constant privacy guarantee could be more beneficial. We appreciate your perspective and will clarify this point in our discussion.
>
> (5) Last but not least, we thank you very much for the careful review. We will fix all the typos.

---

### Official Review · Reviewer_BT5n · 2025-03-15

**Overall Recommendation:** 4

**Summary:**

This paper examines the regret-privacy trade-off for the Gaussian TS algorithm under Gaussian Differential Privacy. By drawing the connection between Gaussian TS and UCB, authors propose the DP-TS-UCB algorithm, which does not need to sample a Gaussian model at each round, the paper achieves a new privacy-regret trade-off that improves upon the previous state-of-the-art results.

**Claims And Evidence:**

Yes. Most claims are well-supported.

**Essential References Not Discussed:**

No

**Experimental Designs Or Analyses:**

Yes

**Methods And Evaluation Criteria:**

Yes, the evaluation criteria(Regret and GDP) are standard and widely applied in most related areas.

**Other Comments Or Suggestions:**

I don't have other comments.

**Other Strengths And Weaknesses:**

Other Strengths:

The authors provide a clear explanation of the intuition behind their algorithm design, which makes the paper easy to follow.

Weakness:

One potential weakness is the lack of lower bound results on such regret-privacy tradeoff.

**Questions For Authors:**

1. While this paper focuses on the frequentist setting, I am curious whether improved results could be achieved by considering Bayesian regret, even under the assumption of a Gaussian prior. Additionally, I wonder how the variance of prior distributions impacts such tradeoff in heterogeneous reward setting, as in non-privacy setting [1].

2. Are there any lower bound results on such regret-privacy tradeoff?

[1] Saha, A., & Kveton, B. (2023). Only pay for what is uncertain: Variance-adaptive thompson sampling. arXiv preprint arXiv:2303.09033.

**Relation To Broader Scientific Literature:**

This paper follows the line of work exploring the privacy-regret tradeoff for bandits with a Gaussian TS, as listed in the related work section and the table. The contributions made by this paper, including the improved tradeoff bound and the principles behind its algorithm design, provide valuable insights into this line of research.

**Theoretical Claims:**

I have not thoroughly reviewed all the proofs and the appendix in full detail, but based on the provided intuition and the outlined proofs, I am inclined to believe that the theoretical claims are correct.

---

> ### Author Rebuttal · Authors · 2025-04-01
>
> Thank you very much for the constructive comments. We address each of your questions as follows.
>
> (1) Thank you very much for referring us to this interesting paper [1]. Under the notion of Bayesian regret and using Gaussian priors, we think it is possible to achieve an improved trade-off between regret and privacy by changing the variances of prior distributions. It is an interesting research direction, but it is out of the scope of our work as we focus on the frequentist regret setting.
>
> (2) Regarding the lower bounds on the regret-privacy trade-off, lower bounds exist for differentially private bandits under the notion of the classical $(\varepsilon, \delta)$-DP [2, 3, 4]. We will discuss those in the related work. Establishing lower bounds for our specific algorithm is an interesting avenue for future work.
>
> [1] Saha, A., & Kveton, B. (2023). Only Pay for What is Uncertain: Variance-adaptive Thompson Sampling. arXiv preprint arXiv:2303.09033.
>
>
> [2]: Roshan Shariff & Or Sheffet,  Differentially Private Contextual Linear Bandits, NeurIPS 2018.
>
> [3]: Achraf Azize & Debabrota Basu, When Privacy Meets Partial Information: A Refined Analysis of Differentially Private Bandits, NeurIPS 2022.
>
> [4]: Siwei Wang & Jun Zhu, Optimal Learning Policies for Differential Privacy in Multi-armed Bandits, Journal of Machine Learning Research 2024.

---

### Official Review · Reviewer_BBHN · 2025-03-16

**Overall Recommendation:** 4

**Summary:**

This paper describes a stochatic MAB algorithm that preserves DP. It uses Thompson sampling with a limited budget of samples per epoch. Once the samples are exhausted within a round it uses the maximum of those samples. This is akin to an upper confidence bound. It also has a  parameter $\alpha$ that can tune the behaviour from privacy-preserving to regret-minimising.

**Claims And Evidence:**

The presentation is ok, and the sketch proofs are readable. The structure of the proofs makes sense.

**Essential References Not Discussed:**

N/A

**Experimental Designs Or Analyses:**

The experimental analysis is only a minor aspect.

**Methods And Evaluation Criteria:**

Standard bandit experiments.

**Other Comments Or Suggestions:**

It seems that $\alpha$ can tune the algorithm only to a limited extent. Is there a way to achieve e.g. a specific fixed level of GDP? At this point, it seems like, for $\alpha = 0$ we get 2.875-GDP,  or (1, 0.7612)-DP. It still is an improvement over TS-G.

**Other Strengths And Weaknesses:**

The writing could be improved.

**Questions For Authors:**

See above. How meaningful is the DP guarantee?

**Relation To Broader Scientific Literature:**

It is of interest to online learning and DP

**Theoretical Claims:**

I checked the sketch proofs and some of the appendix, but I could not really go through all the algebra.

---

> ### Author Rebuttal · Authors · 2025-04-01
>
> Thank you very much for the constructive comments. Regarding the results of GDP guarantees, we would like to clarify that when choosing $\alpha = 0$, the privacy guarantee depends on $T$. In Theorem 4.4, the GDP guarantee is in the order of $ \sqrt{ T^{0.5(1-\alpha)} \ln^{1.5(1-\alpha)}(T)}$. So, only $\alpha=1$ yields a constant GDP guarantee. Therefore, the algorithm is not always $2.875$-GDP if we change the values of $T$, and $\alpha$ does parametrize a privacy-regret tradeoff. For a fixed $T$ and $\alpha$, it is possible to achieve any specific lower level of GDP via scaling the Gaussian variance by a constant larger than one. However, it will result in an increased regret.

---

### Decision · Program_Chairs · 2025-05-01

**Decision:**

Accept (poster)

**Comment:**

The authors propose a private stochastic multi-armed bandit algorithm based on Thompson sampling, which allows trading-off regret bounds and privacy bounds. The reviewers were unanimously positive about the paper and recommended acceptance. We suggest that the authors explain the connection to UCB and the role of the parameter $$\alpha$$ more clearly in the final version.